# Drivers of Antarctic sea ice advance

Kenza Himmich[1] ✉, Martin Vancoppenolle [1], Gurvan Madec[1,2],
Jean-Baptiste Sallée[1], Paul R. Holland[3] & Marion Lebrun[1,4]

Antarctic sea ice is mostly seasonal. While changes in sea ice seasonality have been observed in recent decades, the lack of process understanding remains a key challenge to interpret these changes. To address this knowledge gap, we investigate the processes driving the ice season onset, known as sea ice advance, using remote sensing and in situ observations. Here, we find that seawater freezing predominantly drives advance in the inner seasonal ice zone. By contrast, in an outer band a few degrees wide, advance is due to the import of drifting ice into warmer waters. We show that advance dates are strongly related to the heat stored in the summer ocean mixed layer. This heat is controlled by the timing of sea ice retreat, explaining the tight link between retreat and advance dates. Such a thermodynamic linkage strongly constrains the climatology and interannual variations, albeit with less influence on the latter.

The Antarctic sea ice seasonal wax and wane is one of the most spectacular, climate-related signals, with large consequences for the global ocean water mass structure and circulation[1–3], the Earth's energy budget[4], and marine ecosystems[5,6].

The seasonal cycle of Antarctic sea ice is marked by two key transitions between open water and ice-covered conditions: advance and retreat. Advance or retreat are defined as the first day in the year when smoothed-in-time sea ice concentration exceeds or falls below 15%[7–9]. Over the last 40 years, changes in the timing of Antarctic sea ice advance and retreat have been documented from satellite-based passive microwave sensors[7,10]. The changes, highly variable regionally, were attributed to wind-driven changes in ice transport and seasonal thermodynamic ice-ocean feedbacks[7,11–15]. Yet interpretation is complicated: strong interannual variability dominates the trends in the last two decades[16] and drivers involve multiple oceanic and atmospheric processes[7,10,17,18], in a context of limited understanding of the drivers of sea ice advance and retreat.

Sea ice advance and retreat are influenced by different processes implying specific observational needs and problems. In this paper, we focus on the fundamental drivers of the sea-ice advance date and on links to the sea-ice retreat date. Sea-ice advance can be controlled either by the freezing of seawater or by sea-ice drifting from already frozen areas[19–22]. Freezing starts ultimately once the entire mixed layer,

having warmed up from solar absorption in spring and summer then cooled down through fall, approaches the freezing temperature[23,24]. Close to the winter sea ice edge, freezing can, however, be inhibited by entrained[25] or advected[20] oceanic heat into the mixed layer.

As such, one can hypothesize that the date of sea ice advance is controlled by the upper ocean heat content, surface fluxes, sea ice thermodynamics and drift. To understand these contributions, we relate climatological dates of sea ice advance derived from passive-microwave sea ice concentration[26] to recently available observational datasets. They include passive microwave-based sea ice concentration budget diagnostics that split sea ice changes into dynamic (i.e., drift-related) and thermodynamic (i.e., related to freezing) process contributions[22]; thermal infra-red radiance satellite sea surface temperature[27] (SST); compilations of in situ hydrographic measurements[28] which now provide a detailed climatological view of the upper oceanic thermohaline structure under Antarctic sea ice, thanks to animal-borne sensor records[29]. Combining these sources, we highlight a strong overarching contribution of upper ocean thermodynamics in setting the climatological date of sea ice advance and an important role for ice drift in an outer band, with a width of a few degrees of latitude. We then provide evidence that these mechanisms also contribute to a certain extent to observed interannual changes in the timing of sea ice advance.

[1]Sorbonne Université, Laboratoire d'Océanographie et du Climat, CNRS/IRD/MNHN, Paris, France. [2]Université Grenoble Alpes, Inria, CNRS, Grenoble INP, LJK, 38000 Grenoble, France. [3]British Antarctic Survey, Cambridge, UK. [4]Takuvik, Université de Laval, Québec, QC, Canada.
✉ e-mail: kenza.himmich@locean.ipsl.fr

## Sea ice advance: local freezing or import of remote ice?

The satellite-based sea ice concentration budget, based on sea ice drift and coverage retrievals from AMSR-E products over 2003–2010[11,18], is used to assess how thermodynamic (i.e., freezing of seawater) and dynamic (i.e., import of sea ice) processes control the spatial variability of climatological dates of advance ($d_a$) (Fig. 1a). The sea-ice concentration budget cannot be evaluated prior to $d_a$, when the processes leading to sea ice advance take place, because of missing ice drift data and large sea ice concentration errors in the low-concentration ice near the ice edge. Instead, we evaluate the thermodynamic (Th) and dynamic (Dy) contributions to the total sea ice concentration tendency over the 30 days following $d_a$, as well as their ratio (Dy/Th) (Fig. 2; see Methods), which delineates regions where transport or freezing dominates sea ice concentration changes.

Following sea ice advance, freezing (Th > 0; Fig. 2a) dominates sea ice concentration tendencies (|Dy/Th| < 1; Fig. 2c) in most of the seasonal ice zone except in a circumpolar band close to the sea ice edge where ice import (Dy > 0, Fig. 2b) takes over freezing (|Dy/Th| > 1) or where net melting occurs (Th < 0). This is consistent with previous work, based on sea ice concentration or volume budget decomposition, which showed that the wintertime ice edge is

sustained by ice transport rather than freezing[20,22,30]. We refer to the region south of the |Dy/Th| = 1 contour, as the inner zone and north of this contour, as the outer zone. The latter represents 32% of the seasonal ice zone area. These zones are robust to the choice of the time window over which the budget is integrated following $d_a$, being weakly sensitive to the window size from 15 to 60 days (Supplementary Fig. 1). We retain a time window of 30 days as a compromise between the needs to be close enough to the time of advance and to maximize the number of useable observations (Supplementary Fig. 1).

We find the inner and outer zones hydrographically differ at the time of advance. Indeed, the sea surface temperature at the date of advance ($SST_{da}$), evaluated from an infrared satellite SST climatology (2003–2010)[27], is consistent with our analysis of the sea ice concentration budget (Fig. 2d). First, similar spatial structures are seen, which is remarkable since both sources are independent. In particular, $SST_{da}$ is significantly warmer than the freezing temperature ($T_f$) in the outer zone (median +/− IQR: 0.6 +/− 0.3 °C). Also, the 5% highest values of $SST_{da}$- $T_f$ (i.e., >1°C and higher than the uncertainty of the SST product, Supplementary Fig. 2), are found in or very close to the outer zone contour (Fig. 2d). The median $SST_{da}$- $T_f$ is lower in the inner zone (0.4 +/− 0.2 °C) than in the outer zone, however this difference is not

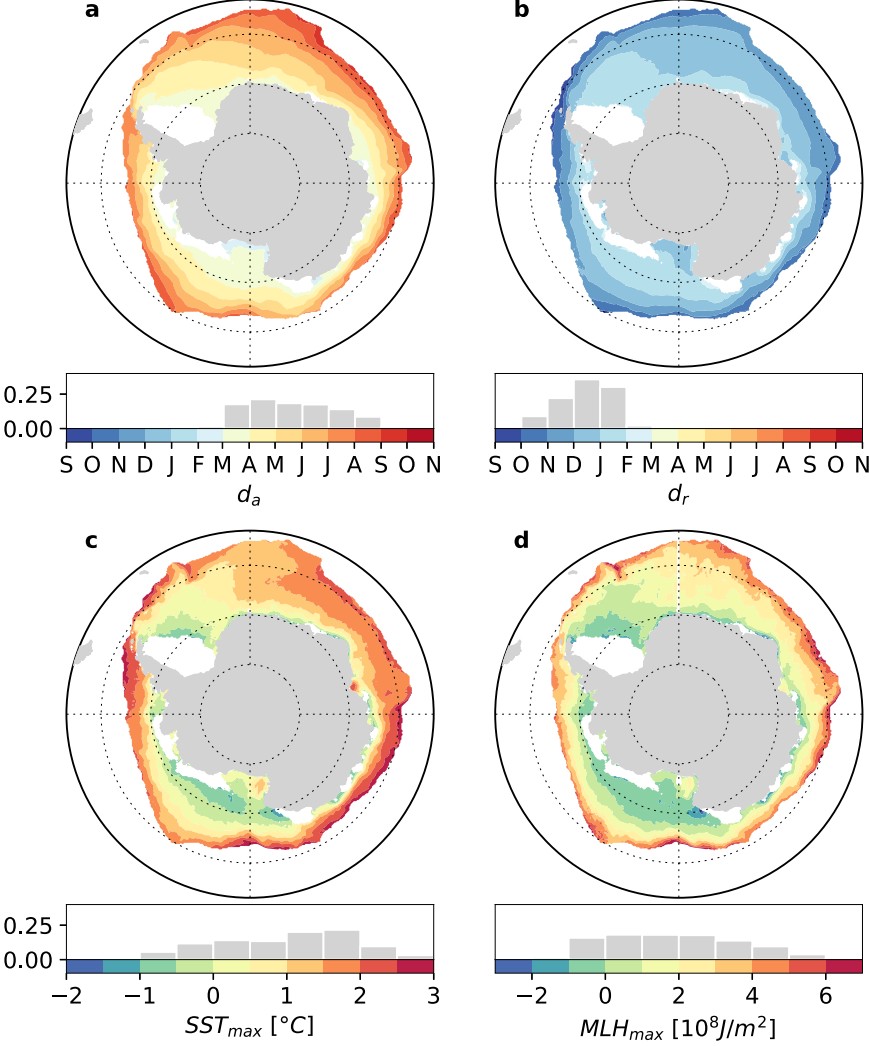

**Fig. 1 | Climatological maps of key variables (1982-2018).** Dates of sea ice **a** advance ($d_a$) and **b** retreat ($d_r$) derived from passive microwave sea ice concentration; seasonal maxima of **c** sea surface temperature ($SST_{max}$) and **d** mixed layer heat content ($MLH_{max}$) from a climatology of thermal infra-red radiance satellite sea surface temperature and a climatology of mixed layer depths, constructed from in situ observations. Corresponding frequency histograms are shown under each map. White patches indicate regions out of the seasonal ice zone. Source data are provided as a Source data file.

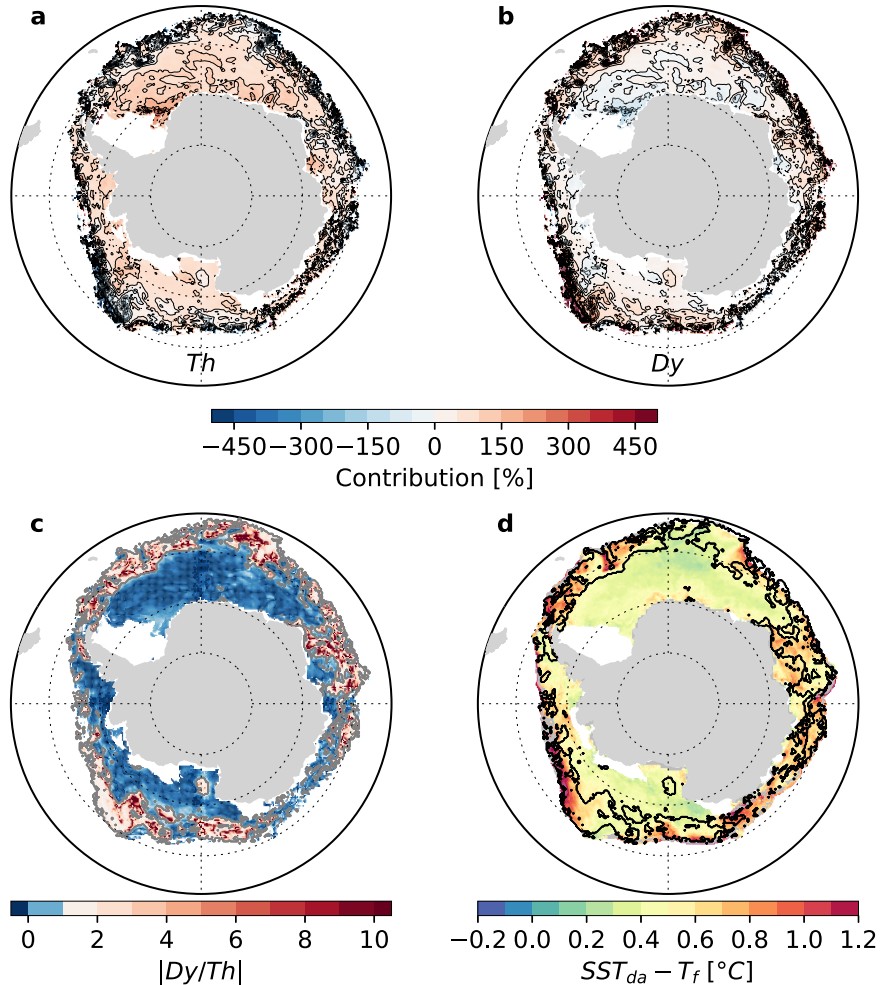

**Fig. 2 | Maps of passive microwave-based sea ice concentration budget terms and infra-red radiance satellite sea surface temperature, near the date of sea ice advance, averaged over 2003–2010.** The thermodynamic (Th, **a**) dynamic (Dy, **b**) contribution to the total sea ice concentration tendency over the 30 days following the date of advance and their absolute ratio ($|Dy/Th|$, **c**), all evaluated over a one-month window following the sea ice advance date. **d** Sea surface temperature at the date of advance referenced to freezing temperature ($SST_{da}$ -$T_f$, with $T_f$ assumed constant at −1.8 °C). Superimposed contour in **d** indicates $|Dy/Th| = 1$ which defines the limit between the inner and outer zones. White patches indicate regions out of the seasonal ice zone and gray patches (in **d**), where the sea ice concentration budget is not defined because of missing ice drift data. Source data are provided as a Source data file.

significant, which may reflect uncertainties in the exact position of the inner-outer zone boundary, or in $SST_{da}$. Nevertheless, these findings are robust to the choice of alternative SST products (satellite[31] and in situ; Supplementary Figs. 2 and 3) and to the choice of a longer considered time period (1982–2018 instead of 2003–2010; Supplementary Fig. 2). Finally, a last element of interest is that the temperature profile at the base of the mixed layer is thermally unstable in the outer zone during the first three months of the advance season, according to an in situ hydrographic climatology[28] (Supplementary Fig. 4). Taken together, we argue that the outer zone corresponds to where drifting ice encounters sufficiently warm waters for net basal melting to occur on the day of advance. The contrast is arguably reinforced by an unstable water column, which could lead to entrainment of warm waters into the mixed layer, opposing sea ice growth. Previous studies have also highlighted the role of oceanic heat supply as a spatial constraint to sea ice advance in the winter ice edge region[20,25].

In conclusion, the sea ice concentration budget, satellite SST and in situ hydrography observations consistently suggest the climatological date of advance in the inner and the outer zones is controlled by different processes. While the onset of freezing controls the date of advance in the inner zone, a more complex balance between ice import

and oceanic heat supply driving basal melting primarily controls the date of advance in the outer zone. We next investigate the physical processes controlling the onset of freezing.

### Control of sea ice advance from ice-ocean thermodynamic processes

In the inner zone, where freezing is the main driver of sea ice advance, we expect the climatological $d_a$ to be strongly linked to the climatological heat content of the mixed layer, as well as the mixed layer cooling rate during the open water season. In this section, we explore the strength of these links. We find that the spatial pattern of $d_a$ relates to the spatial pattern of the seasonal satellite-based[27] SST maximum ($SST_{max}$). Maps of climatological $d_a$ and $SST_{max}$, shown respectively in Fig. 1a and c, indicate that waters with lower seasonal SST maximum freeze earlier. Moreover, a linear model attributes a large part of the spatial variance in $d_a$ to $SST_{max}$ ($R^2 = 0.81$), suggesting that $SST_{max}$ could be a proxy of the mixed layer heat gained in spring and summer, which is then lost before sea ice advance. However, a nonlinearity in the relationship appears when representing the spatial distributions of $d_a$ anomalies versus $SST_{max}$ anomalies on a 2D histogram (Fig. 3a). Using a monthly climatology of mixed layer depths[28], we find that this non-linearity is most

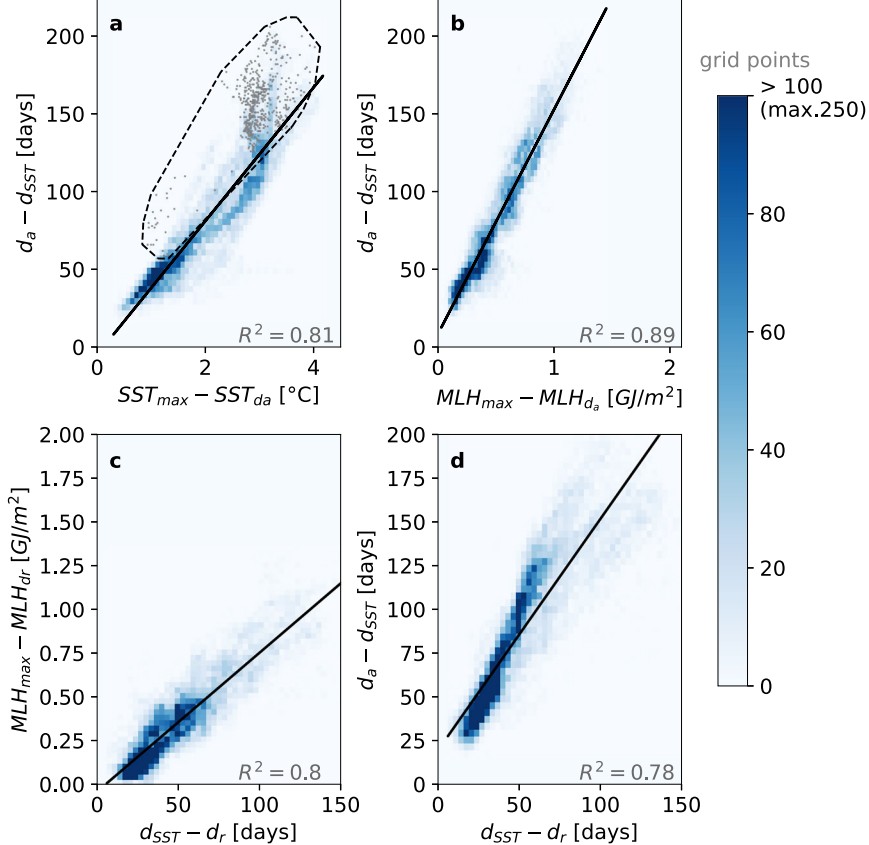

**Fig. 3 | Selected inner zone spatial relationships between the 1982–2018 climatological maps of variables displayed in Fig. 1, plotted as 2D histograms.**
**a** Advance dates ($d_a$) vs seasonal maximum of sea surface temperature (SST$_{max}$), **b** $d_a$ vs seasonal maximum of mixed layer heat content (MLH$_{max}$), **c** MLH$_{max}$ vs retreat dates ($d_r$) and **d** $d_a$ vs $d_r$. Anomalies are used, tailored to best showcase the relevant relationships (see Methods). $d_a$ ($d_r$) anomalies refer to the date of maximum sea surface temperature ($d_{SST}$) such that positive anomalies indicate later advance (retreat). In **b** (**c**), MLH$_{max}$ anomalies refer to the mixed layer heat content value at sea ice advance (retreat) date, which is close to but not exactly zero, because the sea surface temperature is a few tenths of degree above freezing (see Supplementary Figs. 2 and 3). Only grid points from the inner zone were retained. Color gives the number of points in each pixel of the 2D histogram space. In **a**, the black polygon highlights high mixed layer depths, enclosing grid points with a mixed layer deeper than 80 m on average over the open water season; the gray dots refer to the corresponding grid points. A Least Square linear regression was performed for each plot; the corresponding regression line (significant at 99%), and corresponding coefficients of determination ($R^2$) are shown. Source data are provided as a Source data file.

obvious for the deepest mixed layers. In Fig. 3a, the non-linearity is confined in the black polygon enclosing all grid points with an averaged mixed layer depth over the open water season, greater than 80 m. This suggests that the SST insufficiently characterizes the mixed layer heat content (MLH), and that the heat content over the entire mixed layer depth, which itself has a spatial variability, must be considered.

Based on the climatological mixed layer depth and SST$_{max}$, we define an observational estimate of the climatological seasonal maximum of MLH (MLH$_{max}$; see Methods):

$$MLH_{max} \approx \rho c_p MLD_{d_{SST}}.SST_{max} \qquad (1)$$

where MLD$_{d_{SST}}$ is the mixed layer depth evaluated in the month of $d_{SST}$, the climatological date of maximum SST. MLH$_{max}$ accounts for the variability in both SST and mixed layer depth (Fig. 1d). Strikingly, the 2D histogram of $d_a$ anomalies versus MLH$_{max}$ anomalies does not show any evident non-linearity (Fig. 3b). MLH$_{max}$ also explains a larger part of the spatial variance in $d_a$ ($R^2 = 0.89$) than SST$_{max}$ does ($R^2 = 0.81$, Fig. 3a). The observed relationship between MLH$_{max}$ and $d_a$ can be understood in the framework of a mixed layer heat budget model[9] (see Methods). Integrating this budget for each ($x,y$) grid point over the open ocean cooling period, a direct link between $d_a$ and MLH$_{max}$

anomalies arises:

$$d_a(x,y) - d_{SST}(x,y) = \frac{MLH_{max}(x,y) - MLH_{da}(x,y)}{<Q^-(x,y)>} \qquad (2)$$

where:

$$MLH_{da} \approx \rho c_p MLD_{da}.SST_{da}. \qquad (3)$$

where MLD$_{da}$ is the mixed layer depth evaluated in the month of $d_a$. The average net heat loss during the cooling period $<Q->$ sets the rate of mixed layer heat loss between the date of maximum SST and $d_a$. A linear MLH$_{max}$-$d_a$ relationship over the whole seasonal ice zone would then suggest spatially uniform $<Q->$, which seems to hold overall (Fig. 3b). The scatter associated with this relationship indicates that $<Q->$ varies but is equally distributed around the mean, without altering the linearity of the relationship. Thus, the spatial variability of $<Q->$ only has a minor influence on $d_a$ in the inner zone. Applying Eq. (1) to the slope of the MLH$_{max}$-$d_a$ linear regression model (Fig. 3b), we estimate the average net heat loss $<Q->$ to 80 W/m². This number integrates all mixed layer heat budget contributors (entrainment, advection, diffusion and air-sea fluxes) but is likely dominated by air-sea fluxes[32]. Such net air-sea heat loss is consistent with reanalysis-

based estimates of net surface fall heat loss in Antarctic ice-free waters (e.g., ref. 33).

The date of advance in the inner zone is therefore controlled by the heat that accumulates in the mixed layer during the ice-free season. This heat is tightly related to the net radiative energy input at the ocean surface (turbulent fluxes are much weaker than radiative fluxes in the sea-ice zone[34]; Supplementary Fig. 5), which is itself constrained by the presence of sea ice and hence, by the date of sea ice retreat ($d_r$). Consistently, we find a remarkably strong linear link between climatological $d_r$ and $MLH_{max}$ ($R^2 = 0.80$; Fig. 3c). This suggests that $MLH_{max}$ is mostly set by the timing of sea ice retreat and weakly influenced by the spatial variability of net heat fluxes warming the mixed layer during the ice-free season (see Methods). Therefore, by controlling amount of heat accumulating in the mixed layer over the ice-free period, the timing of ice retreat indirectly controls the timing of ice advance. Comparing the climatology of $d_r$ with that of $d_a$ consistently indicates that later $d_r$ is associated with earlier $d_a$, with a significant and strong linear relationship ($R^2 = 0.78$, Fig. 3d). Previous work has already linked interannual anomalies in $d_a$ to anomalies in $d_r$[7]. Here, we show that this link holds for the spatial variability of climatological retreat and advance dates, and is controlled by the upper ocean heat content.

The statistical relationships between $d_r$, $MLH_{max}$, and $d_a$ are also strong in the outer zone, but generally not as much as in the inner zone (Supplementary Fig. 6). The $MLH_{max}$-$d_a$ link is weaker in the outer zone ($R^2 = 0.83$) than in the inner zone ($R^2 = 0.89$), but still explains a large part of the $d_a$ variance. Similarly, the $d_r$-$MLH_{max}$ link is weaker ($R^2 = 0.72$, $p < 0.01$) in the outer zone than in the inner zone ($R^2 = 0.80$). This general weakening and the associated larger regression errors might reflect a larger spatial variability in net heat fluxes in the outer zone (see Methods), possibly linked to the entrainment of warm waters into the mixed layer (Supplementary Fig. 4). The departure from the linear relationship occurs in regions of the outer zone that differ between the $d_r$-$MLH_{max}$ and the $MLH_{max}$-$d_a$ relationships (Supplementary Fig. 7). This spatial mismatch affects the $d_r$-$d_a$ relationship, which is therefore weaker than the two others in the outer zone ($R^2 = 0.61$), and does not explain as much of the $d_a$ variance there.

In summary, the climatological $d_r$ strongly affects the climatological $d_a$ in the inner zone only. By contrast, the climatological $MLH_{max}$ determines the climatological $d_a$ throughout the seasonal ice zone, regardless of the processes (freezing or ice import) increasing sea ice concentration at that time.

### From spatial to interannual variability

Thermodynamic processes in ice-free waters provide strong constraints to the climatological date of advance. Whether such mechanisms also apply at interannual time scales is not straightforward. Stammerjohn et al.[7] disclosed significant correlations between detrended dates of retreat and subsequent advance over 1980–2010. An ice-ocean thermodynamic feedback was hypothesized to explain this link. The same mechanism was also identified in the Arctic[7–9,35]. However, observations and CMIP5 model analyses suggest that thermodynamic processes are less effective at explaining interannual variations than they are for the mean state[8,9]. Based on what precedes, we question to which extent our findings on the mean state can be applied to interannual variations.

We expect the ice-ocean thermodynamic feedback to operate in agreement with our $d_r$-$MLH_{max}$-$d_a$ framework: an earlier retreat on a given year would lead to a higher maximum MLH and a later advance. We examine these links at the interannual time scale, using the SST as a proxy for the MLH, due to the limited spatial coverage of interannual mixed layer depth data. Based on detrended time series over 1982-2018, we find significant and relatively strong negative links between anomalies of $d_r$ and subsequent $SST_{max}$ ($p < 0.05$ and $r = -0.6 +/- 0.2$; Fig. 4a), and positive links between anomalies of

$SST_{max}$ and subsequent $d_a$ ($p < 0.05$ and $r = 0.5 +/- 0.2$; Fig. 4b) in large parts of the seasonal ice zone. As a result of the thermodynamic linkage between $d_r$, $SST_{max}$, and $d_a$, we also find relatively strong correlations between detrended anomalies of sea ice retreat and subsequent advance date ($p < 0.05$ and $r = -0.5 +/- 0.2$; Fig. 4c), consistently with Stammerjohn et al.[7]. However, those correlations are weak or statistically insignificant close to the seasonal ice zone edge and also in the East Antarctic and Maud Rise sectors, which indicates that processes distinct from the ice-ocean feedback are also strongly involved (Fig. 4a–c).

The mean state-based decomposition between an inner and outer zone seems relevant to better constrain the role of ice transport and melt processes at the interannual time scale. To explore this idea, we examine the interannual standard deviation in the date of advance. We find that interannual variability is highest within or close to the outer zone (Fig. 4d). This suggests that high interannual variability in the timing of advance is due to variability in either sea ice drift, which relates to variability in winds[11] or in ocean heat input[20], or both. By contrast, the lower variability in $d_a$ in the inner zone could relate to a more prevalent control of thermodynamics on the date of advance. Spatial patterns of detrended correlations between $d_r$, $SST_{max}$ and $d_a$ are also generally in line with the inner-outer zones decomposition. The largest correlations are found in the inner zone (Fig. 4a–c), consistently with thermodynamic processes driving sea ice advance there. However, one difference with our findings related to mean state is that drift and melt processes may also considerably contribute to interannual variability in the date of advance in the inner zone, as indicated by locally existing weak and low significance $d_r$-$SST_{max}$-$d_a$ correlations there (Fig. 4a, b). For instance, close to Maud Rise, the correlations between $d_r$ and $d_a$ are significant (Fig. 4c) but not between $d_r$ and $SST_{max}$ (Fig. 4a) and between $SST_{max}$ and $d_a$ (Fig. 4b). The effects of oceanic heat entrainment[36,37] and advection[14,15] might be more suited to explain the variability in this region, despite being located in the inner zone. We therefore surmise that the inner-outer zones boundary may not be as clear for interannual variations in the date of advance than it is for the climatology.

Ultimately, the drivers of the spatial variability in the climatological date of advance also contribute to a certain extent to the interannual variability. Nonetheless, heat fluxes and transport processes exert a stronger influence on the timing of advance at the interannual time scale, compared to the mean state. Future work may help to clarify the exact role of such processes.

## Discussion

Our findings progress the understanding of the climatological timing of sea ice advance while providing valuable insights on the drivers of interannual changes. We now discuss their implications regarding long-term Antarctic sea ice changes.

Projected future Antarctic sea ice changes vary widely amongst current climate models[38] because of persistent biases and poorly represented physical processes in climate projections, particularly problematic in the Southern Ocean[39]. Our results can be used to evaluate the model representation of the processes driving sea ice seasonality in the Southern Ocean against observations. Primarily, the inner-outer zone decomposition provides a specific approach to validate the ice concentration budget during the ice advance season. Additionally, an examination of the different relations embedded in the $d_r$-$MLH_{max}$-$d_a$ framework can serve as a robust approach to verify the existence of the thermodynamic control of sea ice advance by the ocean.

Furthermore, our results provide important constraints on future long-term Antarctic sea ice changes. Given how strong the $d_r$-$MLH_{max}$-$d_a$ relationships are in the recent mean state, it can be argued that these will still hold for the future Antarctic sea ice mean state, providing helpful constraints to project long-term future changes. Indeed, the

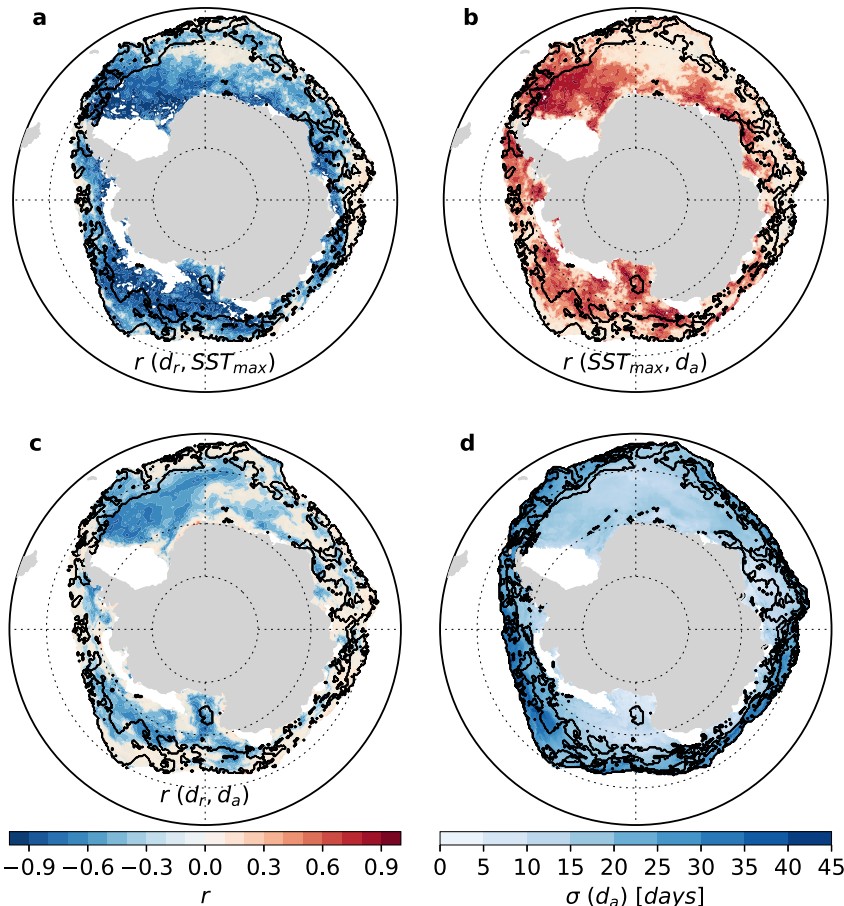

**Fig. 4 | Interannual variability in passive-microwave (1982–2018) date of advance and how it relates to variability in date of retreat and seasonal maximum of sea surface temperature (1982–2018).** Correlation coefficients ($r$) between detrended timeseries of **a** annual dates of retreat ($d_r$) and subsequent seasonal sea surface temperature maximum ($SST_{max}$), **b** annual $SST_{max}$ and subsequent dates of advance ($d_a$), and **c** annual $d_r$ and subsequent $d_a$. **d** Standard deviation ($\sigma$) in the date of advance. Beige shading in **b**–**d** indicates where correlations are not statistically significant at the 95% level. The black contour delimits the inner-outer zone limit derived from the sea ice concentration budget and shown in Fig. 2d. White patches indicate regions out of the seasonal ice zone. Source data are provided as a Source data file.

increased skill of the MLH$_{max}$-$d_a$ relationship, compared to SST$_{max}$-$d_a$ (Fig. 3a, b) emphasizes the importance of considering changes not only in the mixed layer temperature but also in mixed layer depth to fully understand long-term changes. Arguably global warming will be associated with earlier retreat and warmer surface waters, providing more heat to the mixed-layer in summer, delaying sea ice advance. However, changes in mixed layer stratification are also operating and might compete with the effects of the temperature increase. The increase in freshwater input to the subpolar Southern Ocean through increased precipitation[40] and ice sheet mass loss[41], increases the regional upper ocean stratification[28,42,43] potentially reducing the mixed-layer heat content and act against warming by allowing for earlier date of advance, even with an increased surface heat uptake. More work will be needed to understand how temperature and stratification processes drive and respond to long-term sea ice seasonality changes.

## Methods

### Observational data sources
We assess the relationships between the date of sea ice advance and the state of the underlying ocean based on a number of observational data sets. We use daily passive microwave sea ice concentration (SIC) from the EUMETSAT Ocean and Sea Ice Satellite Application Facility[26] (OSI SAF) over 1982–2018 (OSI-450 from January 1982 to April 2015, and OSI-430-b after April 2015). For the sea surface temperature (SST),

we use a daily satellite product available from 1982, based on thermal infra-red radiance measurements, and taken from the global L4 (gap-free, gridded) European Space Agency (ESA) SST Climate Change Initiative (CCI) analysis with a resolution of 0.05°[27], provided with an estimate of the analysis uncertainty on the SST.

We also use a gap-filled monthly 1979–2018 climatology of mixed layer depth and stratification, based on in situ observations[28]. Conductivity–temperature–depth (CTD; 1970–2018), Argo floats (Argo international programme[44]; 2000–2018) and marine mammal-borne sensor profiles (Marine Mammals Exploring the Oceans Pole to Pole programme[45]; 2004–2018) were included. Generalized least squares linear-regressions of individual in situ profiles are performed around each grid point to produce gridded maps of climatological mean fields.

Other datasets are used to support our analysis. To evaluate the radiative heat fluxes during the open water season, we derive a daily climatology of surface shortwave and longwave radiative fluxes from the FH-series data of the International Satellite Cloud Climatology Project[46,47], available over 1982-2016 (ISCCP). Finally, we use NOAA Advanced very High Resolution Radiometer (AVHRR) Optimum Interpolation (OI) 0.25° daily SST v2.0 analysis data[31], also referred to as Reynolds' SST, to ensure the robustness of our analysis.

All data were interpolated on the OSI-SAF Equal-Area Scalable Earth 2 (EASE2) 25 km grid.

## Diagnostics of sea ice and ocean seasonality

Climatological mean dates of sea ice retreat ($d_r$) and sea ice advance ($d_a$) were derived from OSI SAF sea ice concentration, on which we applied a 15-day temporal filter to avoid retaining any $d_a$ or $d_r$ reflecting short events[9]. These dates are defined consistently with previous work[7,9,16,48]. $d_r$ is defined as the first day filtered sea ice concentration drops below 15% while $d_a$ is the first day filtered sea ice concentration exceeds 15%. To ensure $d_r$ and subsequent $d_a$ of the same yearly seasonal cycle are retained, we looked for $d_a$ ($d_r$) starting on a month where no sea ice advance (retreat) occurs, on average over 1982–2018. We selected January 1 of the current year as the start date for $d_a$, and May 1 of the previous year for $d_r$, since the majority (>99%) of $d_a$ and $d_r$ occurs after those dates.

To obtain a meaningful 1982–2018 climatological average of $d_a$ and $d_r$, a missing value is assigned where the number of years with undefined $d_a$ and $d_r$ (corresponding to year-round ice-free or ice-covered grid points) is less than one third of the total number of years in the considered period, following ref. 9.

Other climatological diagnostics were calculated to diagnose the seasonality of upper ocean thermodynamics, using the ESA CCI satellite SST over 1982–2018. For each year, the seasonal maximum of SST, $SST_{max}$ and date when this maximum is reached, $d_{SST}$ were identified during the open water season, between $d_r$ and $d_a$ of the corresponding year. We also calculated the yearly SST on the days of advance ($SST_{da}$) and retreat ($SST_{dr}$). Then, the 1982–2018 average of each of the four ocean seasonality diagnostics was obtained following the same method as for climatological $d_r$ and $d_a$.

## Decomposition of sea ice concentration budget at the time of advance

To explore the respective role of ice dynamics and thermodynamics in setting $d_a$, we evaluate the dynamic and thermodynamic contributions to the sea ice concentration budget at the time of advance. We identify regions of ice import/export, ice melt/growth and regions of dominant dynamic/thermodynamic contributions. We use the sea ice concentration budget decomposition outputs from ref. 22 available at daily frequency between 2003 and 2010. We also use a 2003–2010 climatology of $d_a$, for temporal consistency. These outputs are obtained based on the technique developed by ref. 11 from daily sea ice concentration (NASA Team algorithm[49]) and ice drift fields derived from AMSR-E brightness temperature by a cross-correlation algorithm[50,51]. The governing equation for the sea ice concentration, is decomposed between a dynamic term and a residual:

$$\frac{\partial SIC}{\partial t} = \nabla.(uSIC) + residual \tag{4}$$

The ice concentration flux divergence represents the effects of advection and divergence of sea ice caused by ice drift. The residual term includes both thermodynamic processes (melting/freezing) and mechanical redistribution through ridging and rafting. However, mechanical redistribution should not intervene in the budget at the time of sea ice advance, as it usually occurs for high sea ice concentration. Thus, we consider the residual as purely thermodynamic.

Evaluating the different terms of the previous equation at the time of advance requires analyzing the output of the sea ice concentration budget for sea ice concentration below 15%. However, the budget is not defined at such low sea ice concentration because of missing ice drift data and large sea ice concentration errors near the ice edge. To overcome this limitation, we diagnose total sea ice concentration increase (ΔSIC), as well as percent dynamic (Dy) and thermodynamic (Th) contributions to sea ice concentration tendency during a period on length $\Delta t$ following $d_a$.

The diagnostics are defined as such:

$$\Delta SIC = \int_{d_a}^{d_a + \Delta t} \frac{\partial SIC}{\partial t}\, dt \tag{5}$$

$$Dy = \frac{1}{\Delta SIC} \int_{d_a}^{d_a + \Delta t} \nabla.(uSIC)\, dt \tag{6}$$

$$Th = \frac{1}{\Delta SIC} \int_{d_a}^{d_a + \Delta t} residual\, dt \tag{7}$$

To choose the most suitable upper bound of integration, the sensitivity to $\Delta t$ of the contours delimiting our regions of interest (Th = 0, Dy = 0 and |Dy/Th| = 1) was assessed (Supplementary Fig. 1). For varying $\Delta t$ from 15 to 60 days, we find that the Th = 0 and |Dy/Th| = 1 contours vary only little. More precisely, regions of sea ice melt (Th <0) and dominant dynamic contribution (|Dy/Th| >1) are consistent both in location and percentage of total seasonal ice zone area, strengthening our confidence that they are a close representation of the sea ice concentration budget prior to $d_a$. Hence, our regions of interest should be at similar location and have a similar area at the time of sea ice advance than in any of the time periods $\Delta t$ within the 2 months following $d_a$. We choose $\Delta t = 30$ days as a compromise between a low proportion of missing values in the considered seasonal ice zone and the proximity in time to $d_a$.

## The $d_r$-MLH$_{max}$-$d_a$ relationship in a simple heat budget model framework

The mathematical description of the simple thermodynamic framework used to explain spatial variations in the timing of advance is an updated version of the framework developed by ref. 9 in the context of Arctic sea ice, based on the heat budget in the mixed layer. We define the heat stored in the mixed layer, termed mixed layer heat content (MLH) as:

$$MLH = \rho c_p h T \tag{8}$$

where $h$ is the mixed layer depth, $T$, the mixed layer temperature, $\rho$, the reference density of seawater, and $c_p$, the specific heat of seawater.

The model is based on the temperature balance equation[52], which writes as:

$$\frac{\partial MLH}{\partial t}(t, x, y) = Q_t(t, x, y) \tag{9}$$

with $Q_t$, the total net heat flux in the mixed layer, accounting for surface heat fluxes, entrainment, diffusion and advection. Now, integrating the MLH budget during mixed layer heating (from $d_r$ to $d_{MLH}$) and cooling (from $d_{MLH}$ to $d_a$) periods we get:

$$d_a(x, y) - d_{SST}(x, y) = \frac{MLH_{max}(x, y) - MLH_{da}(x, y)}{<Q^-(x, y)>} \tag{10}$$

$$MLH_{max}(x, y) - MLH_{dr}(x, y) = <Q^+(x, y)>(d_{MLH}(x, y) - d_r(x, y)) \tag{11}$$

where $d_{MLH}$ is the date of maximum MLH. $<Q+>$ and $<Q->$ are respectively the mean total net heat flux during the heating and the cooling periods:

$$<Q^-(x, y)>.\left[d_a(x, y) - d_{MLH}(x, y)\right] = \int_{d_{MLH}}^{d_a} Q_t(t, x, y)dt \tag{12}$$

$$<Q^+(x, y)>.\left[d_{MLH}(x, y) - d_r(x, y)\right] = \int_{d_r}^{d_{MLH}} Q_t(t, x, y)dt \tag{13}$$

Combining Eqs. (12) and (13), we obtain a relationship between $d_r$ and $d_a$:

$$d_a(x,y) - d_{MLH}(x,y) = \frac{<Q^+(x,y)>}{<Q^-(x,y)>} r_{MLH}[(d_{MLH}(x,y) - d_r(x,y))] \quad (14)$$

with:

$$r_{MLH}(x,y) = \frac{MLH_{max} - MLH_{da}}{MLH_{max} - MLH_{dr}}.$$

If $T$ is at the freezing point on $d_r$ and $d_a$, then $MLH_{da} \approx MLH_{dr}$ and $r_{MLH} \approx 1$.

Perfectly linear relationships between climatological $d_a$ and $MLH_{max}$ anomalies, $MLH_{max}$ and $d_r$ anomalies would respectively suggest uniform spatial distributions of $<Q->$ and $<Q+>$. Resultingly, the relationship between $d_a$ and $d_r$ anomalies would also become linear (if $r_{MLH} \approx 1$).

## Definition of the observational MLH

Using the monthly climatology of mixed layer depth from ref. 28 and the ESA CCI SST diagnostics (e.g., $SST_{max}$, $SST_{dr}$, $SST_{da}$), we estimated the observational MLH for any date, $t$, during the open water season as:

$$MLH(t) \approx \rho c_p MLD_t . SST \quad (15)$$

where $MLD_t$ is the monthly mixed layer depth evaluated on the month of the given date, $t$ (e.g., $MLD_{dSST}$ is evaluated on the month of climatological seasonal maximum of SST, $d_{SST}$). The SST is in degrees Celsius. Using this observational estimation of the MLH, we obtain:

$$MLH_{dr} \approx \rho c_p MLD_{dr} . SST_{dr};$$
$$MLH_{max} \approx \rho c_p MLD_{d_{SST}} . SST_{max};$$
$$MLH_{da} \approx \rho c_p MLD_{da} . SST_{da}.$$

## Data availability

The present analyses are mostly based on publicly available observational data. OSI-SAF sea ice concentration data are available from https://osi-saf.eumetsat.int/products. Sea ice concentration budget decomposition outputs are available upon request. Sea surface temperature data are available from https://cds.climate.copernicus.eu/cdsapp#!/home for the ESA CCI product and from https://www.ncei.noaa.gov/products/avhrr-pathfinder-sst for the NOAA AVHRR product. ISCCP radiative surface heat fluxes are available from https://isccp.giss.nasa.gov/projects/flux.html. Climatological fields of mixed layer depth and stratification are available from https://zenodo.org/record/4073174#.YA_jsC2S3XQ. Source data are provided with this paper.

## Code availability

All scripts used for generating the plots in this paper are available from the corresponding author upon request.

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

## Acknowledgements

The authors acknowledge Casimir de Lavergne and François Massonnet for their helpful suggestions. J.-B.S. has received funding from the European Union's Horizon 2020 research and innovation program under grant agreement no. 821001.

## Author contributions

Analysis for this paper was performed by K.H. and supervised by M.V. and G.M. K.H. and M.V. wrote the initial manuscript. G.M., J.-B.S., P.R.H., and M.L. contributed to interpreting the results and improving the paper.

## Competing interests

The authors declare no competing interests.
