## [Peer Review File · Nature Communications]

REVIEWER COMMENTS

Reviewer #1 (Remarks to the Author):

General feedback

1) The presentation in this manuscript is fairly clear and rigorous, with results that add to previous studies of a similar vein. In my estimation the main new results are quantifying the SST near the time of ice advance and showing that the MLH is a slightly better predictor of the advance date than SST is alone in the inner zone and a substantially better predictor in the outer zone. The authors do a good job of citing the literature specific to ice advance and retreat. They might find some additional important previous work in the predictability literature.

Given that the main contributions are quantitative estimates of known relationships and mechanisms, I think this paper could be strengthened if the authors said more about how their results could be used to evaluate and improve CMIP models. What matters most? Is it the size of the inner and outer zones that matter, the correlations or regression coefficients among certain variables, the distributions of the quantities in the 2D histograms, etc. Which are the target metric models should aim for to achieve specific purposes.

2) The speculation about how ice-ocean feedback will be "reinforced" with global warming is too hand wavy in my view. What does it mean to "reinforce". Does this mean to maintain or strengthen the feedback? I don't think global warming would strengthen the feedback due to the competing effects discussed lines 240-243. I recommend removing this speculation about global warming reinforcing the feedback but keep the discussion about competing effects. In addition, it would be useful if the authors could say what the metric of reinforcing a feedback is and how could we observe or model it.

3) I'd like the authors to consider the consequences of their assumptions in more detail. I appreciate that they have tried in various ways to do so already.

The number one issue I have along these lines is knowing whether I believe that a budget 30 days following is a good representation of the budget prior to da. Stating that it doesn't matter if that window is anywhere from 15 to 60 isn't sufficient since all of the windows are still past da. Whether the budget after da is a good representation of the budget before da could be tested in a model with prescribed winds. Alternatively the authors could embrace the budget following the advance date

for what it is, and explain why we should be interested in this time period. For example, we might be interested in the consequences of sea ice advance on the budget, rather than vice versa.

I was not sure if this 30-day shift is used for the other parts of the analysis near the advance date as well.

For example, is the SST in Fig 1d "at the date of advance" as stated at lines 93 and 105, or is it the SST for 30 days following it. I'm just as interested if it is the 30 days following it (maybe even more interested). In any case, please clarify and if it is the 30-days after, then don't say "at".

4) It would also be good to know what error is incurred by assuming no mechanical redistribution during the 30 days following da. A sea ice model could be consulted to estimate this error too. Mechanical redistribution can only reduce SIC, so neglecting it is a systematic bias that leads to an underestimate of sea ice growth.

5) One of the cooler results in my view is the estimate of the mean SST_da in the outer zone, finding that it is quite a bit warmer than -1.8 C. I'm really dying to know the mean SST_da is in the inner zone too, and recommend this number also be included.

Specific feedback

There are many variables used in this paper. A table of them would be nice for the reader (and might help the authors keep track of them too).

line 40-41: This is an odd thing to say: focus on one and seek to understand another.

line 54: When I read this, I wondered why is it desirable for the observations to be from independent sources when they are observations of different things. I think line 105 largely explains why, but I was left hanging all that time!

line 65: I don't like seeing that $T > 0$ close to the date of advance. I suggest considering using H or something else for thermodynamic, rather than T since T as temperature is so ingrained.

line 100: "sea ice transport is mostly set by seawater freezing" sounds odd. Maybe instead say the "sea ice advance is mostly set by seawater freezing".

line 125: The quantity in Fig 1c is labeled SST_da (the caption and line 105 calls it "at the time of advance") but here the text says Fig 1c has SST_max which I would expect to be quite different Is this a mix up or just not explained well?

line 132: I don't know what I'm supposed to see as "standing out clearly" in Fig 3b. Am I supposed to be looking at the polygon in Fig 3a?

Line 144: Is this supposed to refer to Fig 3b?

line 172: MLD_cp is never defined

Methods

The text doesn't say what dataset is used for sea ice drift, other than AMSR-E. Please give a citation to the product. Apologies if one of those citations is it, but none seemed likely.

line 275: filter not filtering

line 279: 10/1982 to 19/1983 is not clear. It sounds like 13 months.

line 282-284: I'm not following this. Why is da or dr ever undefined?

line 313-315: the lower limit of the integral has a erroneous subscript rather than just da

Reviewer #2 (Remarks to the Author):

Review of “Recent decline in Antarctic sea ice cover from 2016 to 2022: Insights from satellite observations, Argo floats, and model reanalysis” by Kshitija et al.

A substantial volume of recent work has focused on identifying the drivers of Antarctic sea ice change over the satellite record. Winds have been shown to be important in driving variability. This interesting study fills an important gap in the knowledge by demonstrating the role of the summer ocean and highlighting the need to understand changes in both the temperature and depth of the mixed layer.

The authors combine remote sensing data products and new in situ observational datasets to investigate the processes modulating the timing of sea ice advance. They demonstrate that the seasonal ice zone can be divided into an “inner” region, just over 2/3 of the total seasonal ice zone, where the freezing ocean sets the sea ice advance, and an “outer zone”, about a third of the total seasonal ice zone, where sea ice advance is a balance between the import of ice and melting. Despite the different processes operating in these zones, the date of sea ice advance is strongly linked to the amount of heat in the mixed layer. This work supports previous studies that have shown the link between the timing of sea ice retreat and the timing of its subsequent advance and, furthermore, demonstrates that the heat storage in the summer ocean mixed layer can explain this correlation.

I enjoyed reading the manuscript and think it would be of interest to a wide variety of researchers as it provides more context on the variability of Antarctic sea ice. The manuscript is well-structured and well-written – I particularly appreciated the mini summaries at the end of each section. The figures are generally of high quality and support the narrative. I have one reservation about the conclusion, which I note below, along with some minor typos.

One of the interesting results from your analysis is that the correlation between the date of retreat and subsequent date of advance, first noted by Stammerjohn et al, 2012, holds for much of the “inner” zone, where the thermodynamics set the date of advance. The “inner” zone in the Indian Ocean sector doesn’t show a significant correlation (Fig 4a). Clearly, the ocean feedback in response to the timing of the retreat doesn’t hold in this region. You show only a minor influence on the spatial variability of Q in the “inner” zone. Does the lack of correlation in this region affect your conclusion that the MLH controls the link between, e.g., early retreat and later advance?

It would be good to be consistent with the arrangement of the letters that label the figures - e.g. in Fig 1, 'b' is top right, whereas in Fig 2, 'b' is bottom left.

Fig 2 - do 'a' and 'b' have the same colorbar?

Line 101: do you mean to reference Fig 2?

Line 105: add 'the' to the end of the line

Line 132: should this refer to fig 3a rather than fig 3b?

Line 142: you might consider referring to Fig 2b.

Line 172: please define MLDCP

Line 278: typo - change "or" to "of"?

Line 280: Is da (dr)1 day 1 of advance and retreat? It might be helpful to define this somewhere.

Line 281: May 1st seems a very safe start date for calculating dr. Jan 1st seems to be a bit closer to the advance - is there ALWAYS retreat in Dec? I suspect so, but maybe you could add the earliest date of advance in the satellite product to satisfy that you always retain the subsequent date of advance?

Reviewer #3 (Remarks to the Author):

This paper provides a quite straightforward, but illustrative examination of the role of upper ocean heat absorption in driving the length of the open water season, and subsequent retreat of sea ice in summer. Much of the analysis is simplified, and only climatological behaviour is examined (perhaps because of the scarceness of data on mixed layer depth for earlier periods). There are two main conclusions. First, it is shown that thermodynamic ice growth dominates the southern regions during ice advance, and dynamics dominate the outer pack. This is not surprising, and consistent with results such as those of Haumann et al (2016), who show that ice growth dominates ice-ocean interactions in the south, and melt in the north. These are not exactly the same, so it's not simply a rehash of old results, but the results of Haumann et al are primarily because of the relative roles of thermodynamics and dynamics. The reasons for the differences (say, between their Figure 3 and your figure 2) would have been interesting to discuss. That said, I do think this is a useful result. The second main result shows that this difference causes a relationship between the date of advance and retreat and the mixed layer heating in the inner zone, and weaker in the outer zone. Again,

perhaps not terribly surprising, as other publications have shown that correlations between advance and retreat dates are stronger for months where ice extent is less (i.e. in the inner zone), and particularly for the date of retreat and the following date of advance. However, while the reasoning has previously been attributed to the ice-ocean heating feedback, it has not been explicitly shown as it is here. So again, this paper shows this nicely in a simple way.

My main criticism (if you can call it that) is not about the correctness of any of the results, but that doing so with climatology only explains a general, not so unexpected behaviour, and so cannot address how much these processes might have contributed (as a feedback) to recent sea ice variability or trends. This is perhaps unfair, as I realize looking at interannual variability and trends was not the author's intention. But as my impression is that the results are not unexpected, the impact will be less.

What would have made the paper more impactful would have been to explore in more detail the interannual and spatial variability. For example, figure 4 seems underexplored. Also, the relationships shown in Fig 3 are strong, but much of the 'interesting' signal is in the deviations from these relationships that likely govern how advance, retreat, and mixed layer heating vary spatially and interannually. I would have appreciated some analysis of how these relationships (and in S3 and S4) might vary spatially, and whether they might have changed over time. There is also no analysis of what may be driving the differences between the inner and outer pack (merely statements about differences in drift and melting). For example, some papers have shown differences in ocean heat flux convergence control differences in the inner and outer zone due to changes in stratification (e.g. Z. Su, GRL, 2017). This would affect the homogeneity of Q , and thus the linearity of your fits. If the stratification in the outer pack is weak, then thermodynamic growth is retarded not so much by remnant summer warming (which may have dispersed over the much longer ice-free period), but by entrainment of ocean heat. Lastly, I think your spatial variability in figure 2b near the ice edge maximum reflects spatial variability in advection, which as noted above will have a very significant impact on your interpretation here because the open water period at the ice edge maximum approaches 12 months.

Nevertheless, the present work does provide a good foundation for exploring these issues in a subsequent paper.

Detailed comments:

1. I suppose you are constrained by space, but I felt like a few more sentences in the introduction providing more insight into the current understanding of causes of observed trends would be helpful, beyond saying "multiple process are involved" with a cite to a 2012 paper. For example,

many papers (some of which you cite) provide some explanation, both atmospheric and oceanic, and associated feedbacks for much of the recent variability. You are quite correct that there is a lot of uncertainty and it is complicated!

2. Line 37 – I think “weak fundamental understanding of the drivers of sea ice advance and retreat” is a bit strong. I think we understand the proximal drivers quite well, but not their relative roles in driving the observed interannual variability.

3. Line 106 “by an infrared satellite SST product

4. Line 105-113. I don’t quite understand how the SST product can be valid in this area for the date of advance, because in figure 2 it appears like you have SST retrievals where you have sea ice cover. The data would not be valid here. Is it from some days before the advance? In any case, this does not seem surprising at all. I am not aware of systematic comparisons with SST, but several papers have noted waters above freezing and their role in delaying advance.

5. Figure 3 – It is not entirely clear to me what data are used here. The caption says 1983-2018 climatological maps, and the text says monthly climatologies, so have you used climatological (i.e. not varying with year) MLDs and annually varying SSTs? Or have you used both annually varying SSTs and MLDs, or something else? If it is annually varying MLDs, then it is highly likely that some of the older period is very data sparse (pre Argo and pre marine mammal data), and that could account for some of the data spread. If either is climatological, then I would not be surprised at all by the spread and then I question how accurate this analysis is. This is perhaps the main reason you might look at only the climatologies – because the MLD data is sparse, and looking at annual variability and trends may be prone to significant errors.

6. Lines 145-150 (see also note below on lines 345-365). If I understand correctly, you are using a relationship between the date of advance, date of SST max, the MLH, and the heat loss. This could be made more clear, as when reading I was confused as to whether you actually estimated the fluxes or not.

7. Given that you are assuming Q is uniform everywhere, then the deviations from linearity are entirely unsurprising in figure 3, and I am not sure you can claim it is due to issues with using SST to describe MLH. Have you checked to see if these deviations are occurring in places where heat fluxes could be unusual?

8. Line 180-182 – the devil is in the details here! Yes, you show that the relationships are broadly linear and so Q is fairly uniform, but this does not mean that the variability in Q is not very important for driving variability in d_a or d_r , and this is really what would be most relevant to recent sea ice variability, and would require a much more in depth analysis. So I don’t think you can say it is weakly influence (the variability appears to be a pretty good number of days, if d_a-d_{sst} is suggestive). It would be nice to know what that variability is compared to the interannual variability of d_a .

9. Line 182-184. I think this is maybe one of the more significant results, and perhaps should be in the main paper. But it is perhaps not surprising that the mixed layer heat increase is driven by the radiative fluxes. It glosses over a lot, of course, and it would be interesting if you could show regional variations. i.e. does it matter more in the Ross Sea vs say, the east Antarctic? I do think you overinterpret the significance a bit. The scatter is not so small, and is due to advective heat and

turbulent fluxes (and also uncertainty, of course) For instance, the turbulent fluxes can be as large as the net longwave flux. In some areas, advection of heat may be very important. And of course, the variance around this line is important in understanding the spatial and temporal variability.

10. Line 189 – I am not clear on how reference 7 was not also making the link on the seasonal scale?

11. Line 199-201. There is some prior work that suggests these links (off the top of my head, I think a Reid et al. paper has a diagram that shows the persistence of anomalies, which for winter and spring would represent the outer region based on ice extent).

12. Also in Methods, you need to provide more information on the time periods for the various data sources for MLD and the gap-filling procedure.

13. Methods – you do not state what data you have used for sea ice drift.

14. Line 341 – “horizontal velocity”

15. Lines 345-365 – I am not 100% sure I understand what you did here. At first, it seemed you computed the mixed layer heat from the total fluxes, but nowhere do you explain how you computed the horizontal advective fluxes, the entrainment flux or the diffusive flux, or what values you use for the various coefficients. So it looks like you don't actually compute fluxes, but estimate if they are important for this broad-scale analysis by the fits in figure 3, and are only deriving a relationship between the flux, the MLH and the date of advance and retreat. This is not very clear, as elsewhere you describe using flux estimates (but for what, I am not sure). If I am correct, this description could be clearer if you simplified by deleting the equation on 338 and start from the equation on 343 (please number your equations!), and just explain that Q_t is the total net flux as you do, and that you are not attempting to estimate it. If I am wrong, and you do compute the fluxes, then you need to explain a lot better.

16. Ref 30 – formatting issue here.

Drivers of Antarctic sea ice advance

Submitted to Nature Communications

Response to reviewers

K. Himmich, M. Vancoppenolle et al.,

June 2023

Reviewer #1

General feedbacks:

1) The presentation in this manuscript is fairly clear and rigorous, with results that add to previous studies of a similar vein.

In my estimation the main new results are quantifying the SST near the time of ice advance and showing that the MLH is a slightly better predictor of the advance date than SST is alone in the inner zone and a substantially better predictor in the outer zone. The authors do a good job of citing the literature specific to ice advance and retreat. They might find some additional important previous work in the predictability literature.

Answer:

Thank you for your comment and your suggestion. Indeed, previous work in the predictability literature supports that anomalies in the date of advance are driven by ice ocean feedbacks.

Actions:

- **We added new references in the introduction where we refer to thermodynamic feedbacks driving sea ice advance:**

Lines 33-34: *"The changes, highly variable regionally, were attributed to wind-driven changes in ice transport and seasonal thermodynamic ice-ocean feedbacks ([...] Bushuk et al 2021; Holland et al 2013)."*

- **We also used those citations to interpret our new Figure 4**

Lines 255-256: *"The effects of oceanic heat entrainment and advection (Bushuk et al 2021; Holland et al 2013) might be more suited to explain the variability of this region, despite being located in the inner zone."*

Given that the main contributions are quantitative estimates of known relationships and mechanisms, I think this paper could be strengthened if the authors said more about how their results could be used to evaluate and improve CMIP models.

What matters most? Is it the size of the inner and outer zones that matter, the correlations or regression coefficients among certain variables, the distributions of the quantities in the 2D histograms, etc.

Which are the target metric models should aim for to achieve specific purposes.

Answer & Action:

You are correct, we were not fully explicit as to how our results could be used to evaluate ocean and sea ice models. However, it is hard to be prescriptive in advance in terms of the tools to be used for model evaluation without trying them.

Nevertheless, two lines of evaluation are possible and **we now refer to them in the main text:**

- Check whether the *inner* and *outer zones* exist and how they are positioned. This would help to evaluate whether a model realistically reproduces the thermodynamic and dynamic processes leading to sea ice advance.
- Check the relations between $SST_{max} / MLH_{max} / dr$ and da against observations (slope and correlation coefficient). This would help to verify the existence of the thermodynamic control of sea ice advance by the ocean ML (and not by something else).

Both evaluations would help to better characterize ocean-ice-atmosphere interaction processes, providing more interesting evaluation than static metrics (i.e. ice extent).

Lines 271-276: *“Our results can be used to evaluate the model representation of the processes driving sea ice seasonality in the Southern Ocean against observations. Primarily, the inner-outer zone decomposition provides a specific approach to validate the ice concentration budget during the ice advance season. Additionally, an examination of the different relations embedded in the $dr-MLH_{max}-da$ framework can serve as a robust approach to verify the existence of the thermodynamic control of sea ice advance by the ocean.”*

2) The speculation about how ice-ocean feedback will be "reinforced" with global warming is too hand-wavy in my view. What does it mean to "reinforce". Does this mean to maintain or strengthen the feedback? I don't think global warming would strengthen the feedback due to the competing effects discussed lines 240-243. I recommend removing this speculation about global warming reinforcing the feedback but keep the discussion about competing effects. In addition, it would be useful if the authors could say what the metric of reinforcing a feedback is and how we could observe or model it.

Answer & Action:

As you pointed out, we only aimed to describe the competing effects of MLD changes and temperature changes on sea ice advance. Phrasing it as a "feedback reinforcement" was indeed not the most suitable way to describe this mechanism. **We reformulated this part of our discussion on long-term changes.**

Lines 283-286: *“Arguably global warming will be associated with earlier retreat and warmer surface waters, providing more heat to the mixed-layer in summer, delaying sea ice advance.”*

However, changes in mixed layer stratification are also operating and might compete with the effects of the temperature increase.”

3) I'd like the authors to consider the consequences of their assumptions in more detail. I appreciate that they have tried in various ways to do so already.

The number one issue I have along these lines is knowing whether I believe that a budget 30 days following is a good representation of the budget prior to da . Stating that it doesn't matter if that window is anywhere from 15 to 60 isn't sufficient since all of the windows are still past da . Whether the budget after da is a good representation of the budget before da could be tested in a model with prescribed winds.

Answer:

You are correct. That the *inner-outer zones* size and location is not sensitive to the time window of calculation after da only shows that our division of the SIZ is robust over the 15 to 60 days following the date of advance (see Figure S1).

However, we find hydrographic evidence that the SIC budget-based *inner* and *outer zones* are a close (but imperfect) representation of the processes occurring prior and at the date of advance: the *outer zone* contour corresponds mostly to the highest SST_{da} (see Figure 2) and the ML is thermally unstable at the start of the advance season in the *outer zone* (see Figure S4).

It is true that this could be tested in a model. However, we would also not be certain if the sea ice concentration budget is well represented in the model and consequently, if the *inner* and *outer zones* are well represented as well.

Actions:

- **We clarified the hydrographic evidences showing that the *inner-outer zones* are indeed representative of the processes occurring before sea ice advance:**

Lines 88-91: *“We find the inner and outer zones hydrographically differ at the time of advance. Indeed, the sea surface temperature at the date of advance (SST_{da}), evaluated from an infrared satellite SST (climatology (2003-2010), is consistent with our analysis of the sea ice concentration budget (Fig. 2d).”*

Lines 99-105: *“Finally, a last element of interest is that the temperature profile at the base of the mixed layer is thermally unstable in the outer zone ($Nt^2 < 0$) during the first three months of the advance season, according to an in situ hydrographic climatology²⁸ (Supplementary Fig. 4). Taken together, we argue that the outer zone corresponds to where drifting ice encounters sufficiently warm waters for net basal melting to occur on the day of advance. The contrast is arguably reinforced by an unstable water column, which could lead to entrainment of warm waters into the mixed layer, opposing sea ice growth. “*

- **We mentioned that there is an uncertainty on the exact *outer zone* contour:**

Lines 95-97: *“The median $SST_{da} - T_f$ is lower in the inner zone (0.4 ± 0.2 °C) than in the outer zone, however this difference is not significant, which may reflect uncertainties in the exact position of the inner-outer zone boundary, or in SST_{da} .”*

Alternatively, the authors could embrace the budget following the advance date for what it is, and explain why we should be interested in this time period. For example, we might be interested in the consequences of sea ice advance on the budget, rather than vice versa.

Answer:

This is possible and we thank you for the suggestion. However, we consider this would imply a wide range of processes unexplored here, and so out of the scope of this paper, as we are interested in what drives sea ice advance dates and not the other way around.

I was not sure if this 30-day shift is used for the other parts of the analysis near the advance date as well.

For example, is the SST in Fig 1d "at the date of advance" as stated at lines 93 and 105, or is it the SST for 30 days following it. I'm just as interested if it is the 30 days following it (maybe even more interested). In any case, please clarify and if it is the 30-days after, then don't say "at".

Answer & Action:

You are right, this was not always clear in the main text. We did write 'at' only to refer to the SST evaluated on the exact day of sea ice advance at lines 89 and 120 (previously 93 and 105). But we were indeed unclear when referring to the budget calculated in the 30 days following *da* when we wrote "close to the date of advance". **We clarified this.**

Line 66: "[...] over the 30 days following da [...]"

Line 69: "Following sea ice advance [...]"

4) It would also be good to know what error is incurred by assuming no mechanical redistribution during the 30 days following *da*. A sea ice model could be consulted to estimate this error too. Mechanical redistribution can only reduce SIC, so neglecting it is a systematic bias that leads to an underestimate of sea ice growth.

Answer:

You are correct, neglecting mechanical redistribution could induce an underestimation of thermodynamic sea ice growth. To ensure our main result (i.e. the *inner-outer zones* contour) is not affected by mechanical redistribution, we identified the regions where such a process is likely to occur over the 30 days following advance, following the method of Holland & Kimura (2016), as shown in the Figure A. The figure shows that mechanical redistribution is only likely to occur in the *inner zone*, which would not be affected by the corresponding bias. Therefore, mechanical redistribution does not affect the main results of our analysis.

Figure A $-|Dy/Th|$ as shown in Figure 2 (of the main text) from the observed SIC budget with corresponding regions where mechanical redistribution possibly occurs delimited by the pink contour

5) One of the cooler results in my view is the estimate of the mean SST_{da} in the outer zone, finding that it is quite a bit warmer than -1.8 C. I'm really dying to know the mean SST_{da} is in the inner zone too, and recommend this number also be included.

Answer:

In the *inner zone*, SST_{da} is also above the freezing point on average by 0.4 ± 0.2 °C, compared to 0.6 ± 0.3 °C in the *outer zone*. On average, SST_{da} is thus not significantly lower in the *inner zone* than it is in the *outer zone*.

However, the frequency of the highest above freezing SST_{da} is higher in the *outer zone* than in the *inner zone* which is what matters the most to support the existence of our *outer zone* (Figure B).

Finally, as noted in the main text and in the Supplementary Materials, the closeness of the average SST_{da} in the two zones can be explained by:

- Errors and spatial averaging effects on the SST product which might increase SST_{da} to a temperature above the freezing point in the *inner zone* (see Figure S2).
- The *outer zone* contour as diagnosed from the budget 30 days following da is not perfect. Some grid points located in the *inner zone* following da might in fact be in the *outer zone* prior to da .

Action:

- **We added the mean SST_{da} in the *inner zone* and we mentioned the possible reasons for the closeness with the average SST_{da} in the *outer zone*: in the main text:**

Line 95-97: “The median SST_{da} is lower in the inner zone (0.4 ± 0.2 °C) than in the outer zone, however this difference is not significant, which may reflect uncertainties in the exact position of the inner-outer zone boundary, or in SST_{da} .”

Figure B – Histograms comparing SST_da in the inner zone (blue) to SST_da in the outer zone (orange)

Specific feedback on results:

There are many variables used in this paper. A table of them would be nice for the reader (and might help the authors keep track of them too).

Answer & Action:

We think this comment might belong to NC editorial policy and will figure that out with the editor at a later stage. **Meanwhile, we use less acronyms and explicitly refer to several variables for more clarity.**

line 40-41: This is an odd thing to say: focus on one and seek to understand another.

Answer & Action:

Thanks for pointing that out. **We changed the phrasing.**

Lines 39-40 *“In this paper, we focus on the fundamental drivers of the sea-ice advance date and on links to the date of sea-ice retreat.”*

line 54: When I read this, I wondered why is it desirable for the observations to be from independent sources when they are observations of different things. I think line 105 largely explains why, but I was left hanging all that time!

Answer & Action:

We think it might not be useful to say this at this early stage in the paper. **We now only mention it on line 91 (formerly 105).**

line 65: I don't like seeing that T>0 close to the date of advance. I suggest considering using H or

something else for thermodynamics, rather than T since T as temperature is so ingrained.

Answer:

It might be confusing, it is true. **We followed your advice and changed T's into Th's (D's were changed into Dy's for consistency).**

line 100: "sea ice transport is mostly set by seawater freezing" sounds odd. Maybe instead say the "sea ice advance is mostly set by seawater freezing".

Answer:

This is a mistake, thank you for spotting it. However, the sentence no longer exists.

line 125: The quantity in Fig 1c is labeled SST_da (the caption and line 105 calls it "at the time of advance") but here the text says Fig 1c has SST_max which I would expect to be quite different Is this a mix up or just not explained well?

Answer & Action:

Fig. 1c shows the seasonal maximum of SST (*SST_max*). The label and caption are consistent with this. Former line 125 also refers to *SST_max*.

However, former line 105 refers to *SST_da* which is shown in Fig. 2d. We forgot to mention the corresponding figure and that might have been confusing. Sorry about that. **We now refer to Fig. 2d when presenting *SST_da*.**

Line 89-91: "[...] (*SSTda*), evaluated from an infrared satellite SST (climatology (2003-2010), is consistent with our analysis of the sea ice concentration budget (Fig. 2d)."

line 132: I don't know what I'm supposed to see as "standing out clearly" in Fig 3b. Am I supposed to be looking at the polygon in Fig 3a?

Answer & Action:

We are indeed referring to the polygon but the text was not explicit enough on that matter. **We now explicitly mention that the nonlinearity is contained in the black polygon.**

Lines 139-140: "*In Fig. 3a, the non-linearity is confined in the black polygon enclosing all grid point points with an averaged MLD over the open water season, greater than 80 m.*"

Line 144: Is this supposed to refer to Fig 3b?

Answer & Action:

Yes, but the reference was not explicit. **We now explicitly refer to Fig. 3b.**

Lines 148-149: "*Strikingly, the 2D histogram of da anomalies versus MLHmax anomalies does not show any evident non-linearity (Fig. 3b).*"

line 172: MLD_cp is never defined

Answer & Action:

Our mistake. MLD_cp was not the right variable name but it was used to refer to the averaged MLD over the open water season. **We now explicitly describe the right variable in the caption.**

Line 176-178: *“In a, the black polygon highlights high mixed layer depths, enclosing grid points with a mixed layer deeper than 80 m on average over the open water season”*

Specific feedback on methods:

The text doesn't say what dataset is used for sea ice drift, other than AMSR-E. Please give a citation to the product. Apologies if one of those citations is it, but none seemed likely.

Answer & Action:

We indeed forgot the citations to the product (Kimura and Wakatsuchi 2011; Kimura et al. 2013). Apologies. **We now give more details on the ice drift data and added the corresponding references in the Methods section.**

Lines 343-344: *“[...] and ice drift fields derived from AMSR-E brightness temperature by a cross-correlation algorithm (Kimura and Wakatsuchi 2011; Kimura et al. 2013)”*

line 275: filter not filtering

Answer & Action:

Thanks for spotting this mistake. **We corrected it, line 318.**

line 279: 10/1982 to 19/1983 is not clear. It sounds like 13 months.

Answer & Action:

You are correct, this sentence was not clear. We also judged it was not really useful to understand our methods. **We removed this sentence and referred to 1982 instead of 1983 as the start of the time period over which most of the study is based.**

line 282-284: I'm not following this. Why is da or dr ever undefined?

Answer & Action:

The dates of advance and retreat are not defined for multi-year ice or open ocean. **We now explain this more explicitly.**

Lines 326-327 *“[...] a missing value is assigned where the number of years with undefined da and dr (corresponding to year-round ice-free or ice-covered grid points) [...]”*

line 313-315: the lower limit of the integral has a erroneous subscript rather than just da

Answer & Action:

Our mistake, that was a typo that **we corrected, lines 361-363.**

-

Reviewer #2

General feedback:

Review of “Recent decline in Antarctic sea ice cover from 2016 to 2022: Insights from satellite observations, Argo floats, and model reanalysis” by Kshitija et al.

A substantial volume of recent work has focused on identifying the drivers of Antarctic sea ice change over the satellite record. Winds have been shown to be important in driving variability. This interesting study fills an important gap in the knowledge by demonstrating the role of the summer ocean and highlighting the need to understand changes in both the temperature and depth of the mixed layer.

The authors combine remote sensing data products and new in situ observational datasets to investigate the processes modulating the timing of sea ice advance. They demonstrate that the seasonal ice zone can be divided into an “inner” region, just over 2/3 of the total seasonal ice zone, where the freezing ocean sets the sea ice advance, and an “outer zone”, about a third of the total seasonal ice zone, where sea ice advance is a balance between the import of ice and melting. Despite the different processes operating in these zones, the date of sea ice advance is strongly linked to the amount of heat in the mixed layer. This work supports previous studies that have shown the link between the timing of sea ice retreat and the timing of its subsequent advance and, furthermore, demonstrates that the heat storage in the summer ocean mixed layer can explain this correlation.

I enjoyed reading the manuscript and think it would be of interest to a wide variety of researchers as it provides more context on the variability of Antarctic sea ice. The manuscript is well-structured and well-written – I particularly appreciated the mini summaries at the end of each section. The figures are generally of high quality and support the narrative. I have one reservation about the conclusion, which I note below, along with some minor typos.

Answer:

Thank you!

One of the interesting results from your analysis is that the correlation between the date of retreat and subsequent date of advance, first noted by Stammerjohn et al, 2012, holds for much of the “inner” zone, where the thermodynamics set the date of advance. The “inner” zone in the Indian Ocean sector doesn’t show a significant correlation (Fig 4a). Clearly, the ocean feedback in response to the timing of the retreat doesn’t hold in this region. You show only a minor influence on the spatial variability of Q in the “inner” zone. Does the lack of correlation in this region affect your conclusion that the MLH controls the link between, e.g., early retreat and later advance?

Answer:

The answer is no, because the key result you refer to relates to the climatological date of advance. Figure 4 is based on detrended inter-annual anomalies and hence, relates to interannual variability.

Drivers of the climatological date of advance are not exactly the same as the driver of inter-annual variability. Variability in heat fluxes and transport processes appear to have a larger role to play at the interannual time scale.

Actions:

- **We have rewritten the section “From spatial to interannual variability” (formerly “From spatial to temporal variability”) to better highlight that the climatology and interannual variations are not precisely driven by the same processes (Lines 220-262).**
- **We also explicitly specify that in the Abstract and the Introduction.**

Lines 23-24: *“Such thermodynamic linkages strongly constrain the climatology and interannual variations, albeit with less influence on the latter.”*

Lines 55-57: *“We then provide evidence that these mechanisms also contribute to a certain extent to observed interannual changes in the timing of sea ice advance.”*

Specific feedbacks:

It would be good to be consistent with the arrangement of the letters that label the figures - e.g. in Fig 1, ‘b’ is top right, whereas in Fig 2, ‘b’ is bottom left.

Answer & Action:

Thanks for pointing that out. **We followed your advice to be more consistent.**

Fig 2 - do ‘a’ and ‘b’ have the same colorbar?

Answer & Action:

Yes. **We changed the position of the colorbar for more clarity**

Line 101: do you mean to reference Fig 2?

Answer:

You are correct. However, the sentence no longer exists.

Line 105: add 'the' to the end of the line

Answer & Action:

Thanks for spotting the mistake. **We modified the sentence.**

Line 88-90: “Indeed, the sea surface temperature at the date of advance (*SST_{da}*), evaluated from an infrared satellite SST climatology (2003-2010)”

Line 132: should this refer to fig 3a rather than fig 3b?

Answer & Action:

Yes. Our mistake again. Apologies. **We now refer to the right figure, line 137.**

Line 142: you might consider referring to Fig 2b.

Answer & Action:

Do you mean Fig. 3b? **We now explicitly refer to Fig. 3b in line 149.**

Line 172: please define MLDCP

Answer & Action:

Our mistake. MLD_cp was not the right variable name but it was used to refer to the averaged MLD over the open water season. **We now explicitly describe the right variable in the caption.**

Line 139-140 “the black polygon encloses grid points with an averaged MLD over the open water season above 80 m”

Line 278: typo - change "or" to "of"?

Answer:

Thanks for spotting the typo, however the sentence no longer exists.

Line 280: Is *da* (*dr*)1 day 1 of advance and retreat? It might be helpful to define this somewhere.

Answer:

We do consider the date of advance and retreat as the respective first days of the advance and retreat seasons. Based on observations, the start of the advance and retreat seasons typically corresponds to a threshold sea ice concentration of 15%. Accordingly, in the text we refer to *da* (*dr*) as the first day sea ice concentration exceeds (drops below) 15% (lines 319-321).

Line 281: May 1st seems a very safe start date for calculating *dr*. Jan 1st seems to be a bit closer to the advance - is there ALWAYS retreat in Dec? I suspect so, but maybe you could add the earliest date of advance in the satellite product to satisfy that you always retain the subsequent date of advance?

Answer & Action:

We chose May 1st for *dr* and January 1st for *da* because more than 99% of *dr* and *da* occur after those dates. **We clarified in the Methods that the majority (99%) of *da* and *dr* occur after the chosen start dates.**

Line 323-324: *"We selected January 1st of the current year as the start date for *da*, and May 1st of the previous year for *dr*, since the majority (> 99%) of *da* and *dr* occurs after those dates."*

Reviewer #3

General feedback:

This paper provides a quite straightforward, but illustrative examination of the role of upper ocean heat absorption in driving the length of the open water season, and subsequent retreat of sea ice in summer.

Much of the analysis is simplified, and only climatological behaviour is examined (perhaps because of the scarceness of data on mixed layer depth for earlier periods).

There are two main conclusions. First, it is shown that thermodynamic ice growth dominates the southern regions during ice advance, and dynamics dominate the outer pack. This is not surprising, and consistent with results such as those of Haumann et al (2016), who show that ice growth dominates ice-ocean interactions in the south, and melt in the north. These are not exactly the same, so it's not simply a rehash of old results, but the results of Haumann et al are primarily because of the relative roles of thermodynamics and dynamics. The reasons for the differences (say, between their Figure 3 and your figure 2) would have been interesting to discuss. That said, I do think this is a useful result.

Answer & Action:

We reckon you mean Figure 4c of Hauman et al. (2016) which shows the mean annual ratio of freshwater flux due to dynamic processes (export / import) over the freshwater flux due to thermodynamic processes (freezing or melting).

There is indeed a link with our Figure 2c. The 2 figures are shown below, in Figure C. You are correct, our decomposition is consistent with theirs, at a particular moment of the sea ice seasonal cycle (at the time of advance).

However, there are some methodical differences which would make further comparison too speculative:

- They use a volume budget whereas we use a concentration budget.
- Our budget is strictly observation-based whereas they use a model-based sea ice thickness.

We now refer to Haumann et al.

Lines 72-74 "This is consistent with previous work, based on sea ice concentration or volume budget decomposition, which showed that the wintertime ice edge is sustained by ice transport rather than freezing ([...] Haumann et al., 2016)."

Figure C – Left to right: Figure 4c of Hauman et al. (2016) and Figure 2c of the main text

The second main result shows that this difference causes a relationship between the date of advance and retreat and the mixed layer heating in the inner zone, and weaker in the outer zone. Again, perhaps not terribly surprising, as other publications have shown that correlations between advance and retreat dates are stronger for months where ice extent is less (i.e. in the inner zone), and particularly for the date of retreat and the following date of advance. However, while the reasoning has previously been attributed to the ice-ocean heating feedback, it has not been explicitly shown as it is here. So again, this paper shows this nicely in a simple way.

My main criticism (if you can call it that) is not about the correctness of any of the results, but that doing so with climatology only explains a general, not so unexpected behaviour, and so cannot address how much these processes might have contributed (as a feedback) to recent sea ice variability or trends. This is perhaps unfair, as I realize looking at interannual variability and trends was not the author's intention. But as my impression is that the results are not unexpected, the impact will be less.

Answer:

We are convinced that climatology must be understood before internal variability, and highlight a very strong ($R^2 = 0.9$) relationship in the climatology. We think that is an impactful result because many factors (air-sea fluxes, transport, entrainment) could have weakened the relationships at stake. It is also an impactful result because it provides an important constraint on future long-term Antarctic sea ice changes as explained in the *Discussion* section.

The question of variability is wider because those factors have more influence on the date of advance at the interannual timescale, as we now emphasize in our last *Results* section (“*From spatial to interannual variability*”).

We are currently working on a separate study focusing on the question of the recent variability in sea ice seasonality.

Actions:

- We have rewritten the section “*From spatial to interannual variability*” (formerly “*From spatial to temporal variability*”) (Lines 220-262).

- We also explicitly specify that there are differences between drivers of the mean state and interannual variations in the *Abstract* and the *Introduction*.

e.g. Lines 23-24: *“Such thermodynamic linkages strongly constrain the climatology and interannual variations, albeit with less influence on the latter.”*

Lines 55-57: *“We then provide evidence that these mechanisms also contribute to a certain extent to observed interannual changes in the timing of sea ice advance.”*

- We now better explain the implications on long-term changes of progressing the understanding of the mean state in a separate *Discussion* section.

e.g. lines 278-281: *“Furthermore, our results provide important constraints on future long-term Antarctic sea ice changes. Given how strong the dr - MLH_{max} - da relationships are in the recent mean state, it can be argued that these will still hold for the future Antarctic sea ice mean state, providing helpful constraints to project long-term future changes. Indeed, the increased skill of the MLH_{max} - da relationship, compared to SST_{max} - da (Figs. 3a and b) emphasizes the importance of considering changes not only in the mixed layer temperature but also in mixed layer depth to fully understand long-term changes.”*

What would have made the paper more impactful would have been to explore in more detail the interannual and spatial variability. For example, figure 4 seems underexplored.

Answer:

Thank you for the suggestion. We did expand our analysis of interannual variability to a reasonable extent. We added some panels to Figure 4 to show more explicitly how the thermodynamic processes driving spatial variability of the mean state also contribute to driving interannual variability in the date of advance. We also discussed a bit more the regional patterns.

However, going too far is out of scope: we estimated that a more thorough analysis on variability and trends is out of the scope of this study, because the drivers differ.

Actions:

- We modified our Figure 4 and rewrote the corresponding analysis (Lines 220-262).
- We now explore more in details the regional patterns of Figure 4

Lines 253-257 *“However, one difference with our findings related to mean state is that drift and melt processes may also considerably contribute to interannual variability in the date of advance in the inner zone, as indicated by locally existing weak and low significance dr - SST_{max} - da correlations there (Figs. 4a and b). For instance, close to Maud Rise, the correlations between dr and da are significant (Fig. 4c) but the ones between dr and SST_{max} (Fig. 4a) and SST_{max} and da are not (Fig. 4b). The effects of oceanic heat entrainment and advection might be more suited to explain the variability in this region, despite being located in the inner zone.”*

New Figure 4 of the main text

Interannual variability in passive-microwave (OSI-SAF, 1982-2018) date of advance and how it relates to variability in date of retreat and seasonal maximum SST (CCI, 1982-2018). a Standard deviation in the date of advance, b correlation coefficient between detrended timeseries of dates of retreat and subsequent dates of advance, c of dates of retreat and subsequent seasonal SST maximum and d, of seasonal SST maximum and subsequent dates of advance. Beige shading b, c, and d indicates where correlations are not statistically significant at the 95% level. The black contour delimits the same inner-outer zone limit derived from the SIC budget and shown in Fig. 2d. White patches indicate regions out of the seasonal ice zone.

Also, the relationships shown in Fig 3 are strong, but much of the ‘interesting’ signal is in the deviations from these relationships that likely govern how advance, retreat, and mixed layer heating vary spatially and inter-annually.

Answer:

We agree that interesting signals come from deviations from the linear relationship. We had studied in detail those deviations, as a preliminary analysis.

We thought we could better understand the spatial variability of $\langle Q^+ \rangle$ and $\langle Q^- \rangle$ from the spatial distribution of errors to the relationship (see below Figure D). However, the spatial patterns could not be interpreted without having access to the entire observational heat budget in the ML. This will be done in a model in a following study.

Figure D – Spatial distribution of errors on the dr - MLH_{max} - da linear regressions. Errors are defined as the residuals of a, the da vs MLH_{max} and b, the MLH_{max} vs dr linear regressions. Residuals are calculated as the difference between the actual value of a given grid point and the predicted value based on the model fit.

I would have appreciated some analysis of how these relationships (and in S3 and S4) might vary spatially, and whether they might have changed over time.

Answer & Action:

We tested how these relationships varied (a) regionally (according to the regions of Parkinson et al., (2012); see upper Figure E) and (b) inter-annually (see lower Figure E).

- (a) Overall, our relationships still hold regionally.
- (b) We could not assess how the relationships involving the MLH vary over time because of sparse interannual MLD data. Instead, we used the seasonal maximum SST as a proxy of the maximum MLH and assessed the strength of our relationships for each individual year:
 - R^2 is on average lower when calculated on individual years than when calculated based on the climatology. We think there might be some compensation of spatial anomalies in heat fluxes when calculating the climatology which smooths their spatial distribution and strengthens the linearity of the relationships.
 - However, R^2 is still high for the $dr / SST_{max} / da$ relationships, on average.
 - It is substantially weaker for the dr / da relationship, but this is consistent with our previous analysis (see lines 208-2010; and Figure S6): *“The departure from the linear relationship occurs in regions of the outer zone that differ between the dr - MLH_{max} and the MLH_{max} - da relationships (Fig. S6). This spatial mismatch affects the dr - da relationship, which is therefore weaker than the two others in the outer zone ($R^2=0.61$), and does not explain as much of the da variance there.”*

These results again show that the main drivers of climatological dates of advance have a weaker influence at the interannual timescale, possibly due to a larger role of variability in heat fluxes and transport processes. **We now better emphasize that in the section “From spatial to interannual variability” (Lines 220-262).**

Figure E – Up, Interannual variability and down, regional variability of R^2 from the regressions of Figure 3, S3,S4. Colored dots are yearly values. Colored lines and regional values. Green is for the *inner zone*, red for the *outer zone*. Black dots are the average of yearly or regional values. Blue stars are the climatological values given in Figure 3, S3, S4.

There is also no analysis of what may be driving the differences between the inner and outer pack (merely statements about differences in drift and melting). For example, some papers have shown differences in ocean heat flux convergence control differences in the inner and outer zone due to changes in stratification (e.g. Z. Su, GRL, 2017). This would affect the homogeneity of Q , and thus the linearity of your fits. If the stratification in the outer pack is weak, then thermodynamic growth is retarded not so much by remnant summer warming (which may have dispersed over the much longer ice-free period), but by entrainment of ocean heat.

Answer:

Thank you for the suggestion.

Following your suggestion, we evaluated the stratification at the base of the ML during the advance season to assess the possible role of stratification and entrainment in defining the *outer zone*.

We find that during the first months of the advance season, the temperature profile is unstable at the base of the ML in the *outer zone*, indicating that entrainment of warm water possibly occurs in the *outer zone* at that time. This could possibly prevent freezing from occurring and delay sea ice advance, as you pointed out

However, we are again limited by the available observation of the heat fluxes in the ML: the exact contribution of entrainment to the excess of heat at the time of advance cannot be clearly quantified without having access to a complete heat budget.

Action:

- **We explicitly emphasize that the *inner* and *outer* zones are hydrographically different.**

Line 88 *"We find the inner and outer zones hydrographically differ at the time of advance."*

- **We refer to the possible delay of sea ice advance due to entrainment in the *outer zone*, with reference to Z. Su, GRL, 2017.**

Line 99-105 *"Finally, a last element of interest is that the temperature profile at the base of the mixed layer is thermally unstable in the outer zone ($Nt^2 < 0$) during the first three months of the advance season, according to the in situ hydrographic climatology (Supplementary Fig. 4). Taken together, we argue that the outer zone corresponds to where drifting ice encounters sufficiently warm waters for net basal melting to occur on the day of advance. The contrast is arguably reinforced by an unstable water column, which could lead to entrainment of warm waters into the mixed layer, opposing sea ice growth. Previous studies have also highlighted the role of oceanic heat supply as a spatial constraint to sea ice advance in the winter ice edge region (Su, 2017; Bitz et al., 2005)."*

- **We refer to this as a possible cause for the weakening of the relationships of Fig. 2 in the *outer zone*.**

Line 201-202 *"This general weakening and the associated larger regression errors might reflect a larger spatial variability in net heat fluxes in the outer zone (see Methods), possibly linked to the entrainment of warm waters into the mixed layer (Supplementary Fig. 4)."*

- **We refer to previous literature linked to this in the Introduction.**

Line 43-44 *"Close to the winter sea ice edge, freezing can, however, be inhibited by entrained (Su, 2017) or advected (Bitz et al., 2005) oceanic heat into the mixed layer."*

- **We added the figure below in the Supplementary Materials.**

New Figure S4

Monthly temperature contribution to the stratification at the base of the ML (Nt^2). Maps are climatological, and constructed from in situ observations over 1979-2018. Red contours define the monthly climatology (over 1982-2018) of sea ice extent of the corresponding month, derived from passive microwave data. The black contour defines the limit between the inner and outer zones.

Lastly, I think your spatial variability in figure 2b near the ice edge maximum reflects spatial variability in advection, which as noted above will have a very significant impact on your interpretation here because the open water period at the ice edge maximum approaches 12 months.

Answer:

We understand that you refer to rapid spatial changes in date of retreat near the ice edge and attribute them to advection. This seems indeed likely as these regions are in the outer zone (tailored for the date of advance). However, variations in date of retreat are out of the scope of the manuscript, therefore, we did not modify the manuscript. As a side note, our filtering SIC effectively caps the ice-free season values smaller than 15 days.

Nevertheless, the present work does provide a good foundation for exploring these issues in a subsequent paper.

Answer:

Thank you! We are indeed working on these issues right now.

Specific feedbacks:

1. I suppose you are constrained by space, but I felt like a few more sentences in the introduction providing more insight into the current understanding of causes of observed trends would be helpful, beyond saying “multiple processes are involved” with a cite to a 2012 paper. For example, many papers (some of which you cite) provide some explanation, both atmospheric and oceanic, and associated feedbacks for much of the recent variability. You are quite correct that there is a lot of uncertainty and it is complicated!

Answer & Action:

Stammerjohn et al. (2012) is the last main reference focusing specifically on drivers of changes in the dates of Antarctic sea ice advance and retreat. However, you are correct that there are many references discussing more generally the recent changes in Antarctic sea ice that we did not include there. **We added a few more recent references in a sentence finalizing the paragraph.**

Line 34-37 “*Yet interpretation is complicated: strong interannual variability dominates the trends in the last two decades and drivers involve multiple oceanic and atmospheric processes ([...], Meehl et al., 2019; Eayrs et al., 2021), in a context of limited understanding of the drivers of sea ice advance and retreat.*”

2. Line 37 – I think “weak fundamental understanding of the drivers of sea ice advance and retreat” is a bit strong. I think we understand the proximal drivers quite well, but not their relative roles in driving the observed interannual variability.

Answer & Action:

With this sentence we meant to indicate the lack of studies focusing specifically on the fundamental drivers of the dates of advance and retreat in the Southern Ocean.

But you are correct, our wording is too strong because we have some understanding, especially from previous studies on the seasonal cycle of sea ice extent. **We now use “limited” instead of “weak” (line 37).**

3. Line 106 “by an infrared satellite SST product

Answer:

Thanks for spotting the mistake. The sentence has changed in the modified manuscript.

Lines 89-90 “[...] *evaluated from an infrared satellite SST climatology (2003-2010) [...]*”

4. Line 105-113. I don’t quite understand how the SST product can be valid in this area for the date of advance, because in figure 2 it appears like you have SST retrievals where you have sea ice cover. The data would not be valid here. Is it from some days before the advance? In any case, this does not seem surprising at all. I am not aware of systematic comparisons with SST, but several papers have noted waters above freezing and their role in delaying advance.

Answer:

Both interpolated SST products that we use (ESA CCI & Reynolds; see Methods) provide data at low ice concentration and keep those after quality control. However, you are definitely correct that these can be deemed largely uncertain at the date of advance.

Being aware of that is why we analyze these two fairly different products of SST (and not just one). All products give a consistent picture of larger SST_{da-Tf} in the *outer zone*, despite uncertainties of SST_{da} . Please refer to Figure S2.

Note that we also explored a distribution of *in situ* SST (but there are only a few data points near the date of advance). We find that the distribution of *in situ* SST on the day of advance is in line with the satellite SST on the day of advance (see below Figure).

Actions:

- We added the figure below in the Supplementary Materials.
- We referred to the newly added Supplementary Figure in the main text.

Line 97-98 “Nevertheless, these findings are robust to the choice of alternative SST products (satellite and *in situ*; Supplementary Figs. 2 and 3)”

New Figure S3

Comparison of satellite and *in situ* SST on the day of advance. Frequency histograms of SST at the date of advance (SST_{da}) referenced to freezing temperature (T_f , assumed constant at -1.8°C), derived from satellite (ESA CCI) and from *in situ* SST.

5. Figure 3 – It is not entirely clear to me what data are used here. The caption says 1983-2018 climatological maps, and the text says monthly climatologies, so have you used climatological (i.e. not varying with year) MLDs and annually varying SSTs? Or have you used both annually varying SSTs and MLDs, or something else? If it is annually varying MLDs, then it is highly likely that some of the older period is very data sparse (pre Argo and pre marine mammal data), and that could account for some of the data spread. If either is climatological, then I would not be surprised at all by the spread and then I question how accurate this analysis is. This is perhaps the main reason you might look at only the climatologies – because the MLD data is sparse, and looking at annual variability and trends may be prone to significant errors.

Answer:

In our Figure 3, we only use climatological data for each variable so our analysis is consistent. This is explicitly explained in line 147-148: “Based on the climatological MLD and SST_{max} , we define an observational estimate of the climatological seasonal maximum of MLH”.

You are right, the choice of using only climatological data was driven by the sparseness of MLD at the interannual timescale.

6. Lines 145-150 (see also note below on lines 345-365). If I understand correctly, you are using a relationship between the date of advance, date of SST max, the MLH, and the heat loss. This could be made more clear, as when reading I was confused as to whether you actually estimated the fluxes or not.

Answer & Action:

Thank you for this comment, we understood that the way we estimated the mean flux ($\langle Q \rangle$) was unclear. **We added a sentence to clarify how we estimate mean flux.**

Line 162-163 *“Applying equation (1) to the slope of the MLH_{max} - da linear regression model (Fig. 3b), we estimate the average net heat loss $\langle Q \rangle$ to 80 W/m². ”*

7. Given that you are assuming Q is uniform everywhere, then the deviations from linearity are entirely unsurprising in figure 3, and I am not sure you can claim it is due to issues with using SST to describe MLH. Have you checked to see if these deviations are occurring in places where heat fluxes could be unusual?

Answer & Action:

The uniformity of Q is not an assumption, it is a possible consequence of the linearity of the MLH_{max} - da relationship of Figure 3b. We realize it might not have been clear enough in the Methods (and in the main text, as stated in the previous answer).

As for the deviation from linearity, they could indeed point to specific spatial patterns of Q (see Figure D). However, we do not have good observations of Q to check this (Q is the **total** heat flux in the ML during the cooling period). We plan on working on this question using a model in the future.

We clarified that uniform Q is a consequence of the linearity of the MLH_{max} - da relationship in the Methods section.

Lines 400-401 *“Perfectly linear relationships between climatological da and MLH_{max} anomalies, MLH_{max} and dr anomalies would respectively suggest uniform spatial distributions of $\langle Q \rangle$ and $\langle Q \rangle$. ”*

8. Line 180-182 – the devil is in the details here! Yes, you show that the relationships are broadly linear and so Q is fairly uniform, but this does not mean that the variability in Q is not very important for driving variability in da or dr , and this is really what would be most relevant to recent sea ice variability, and would require a much more in depth analysis. So I don't think you can say it is weakly influenced (the variability appears to be a pretty good number of days, if da - $dsst$ is suggestive). It would be nice to know what that variability is compared to the interannual variability of da .

Answer & Action:

Drivers of mean state differ from drivers of interannual variability. Here in the context of the paragraph, we were discussing the mean state (based on climatological variables). The revised version makes that clearer.

As we mentioned before, the scope of our paper is mostly on mean state, not on variability.

We now explicitly specify that the drivers of mean state and interannual variability are not exactly the same.

e.g. Lines 260-262: “Nonetheless, heat fluxes and transport processes exert a stronger influence on the timing of advance at the interannual time scale, compared to the mean state. Future work may help to clarify the exact role of such processes.”

9. Line 182-184. I think this is maybe one of the more significant results, and perhaps should be in the main paper. But it is perhaps not surprising that the mixed layer heat increase is driven by the radiative fluxes. It glosses over a lot, of course, and it would be interesting if you could show regional variations. i.e. does it matter more in the Ross Sea vs say, the east Antarctic? I do think you overinterpret the significance a bit. The scatter is not so small, and is due to advective heat and turbulent fluxes (and also uncertainty, of course) For instance, the turbulent fluxes can be as large as the net longwave flux. In some areas, advection of heat may be very important. And of course, the variance around this line is important in understanding the spatial and temporal variability.

Answer:

This is indeed a significant result. But we think it is not useful to the main questions of this paper to go into further analyses of this result.

We did assess how this relationship varies spatially and found that it remains strong regardless of the region (see Figure E).

10. Line 189 – I am not clear on how reference 7 was not also making the link on the seasonal scale?

Answer & Action:

Thanks for pointing that out. The “seasonal scale” was not the right formulation for what we aimed to say. **We reformulated the corresponding sentence.**

Line 193-195 *“Here, we show that this link holds for the spatial variability of climatological retreat and advance dates, and is controlled by the upper ocean heat content.”*

11. Line 199-201. There is some prior work that suggests these links (off the top of my head, I think a Reid et al. paper has a diagram that shows the persistence of anomalies, which for winter and spring would represent the outer region based on ice extent).

Answer & Action:

Apologies but we can't seem to find the reference you are referring to. **Note that we refer to other references on predictability of the sea ice edge in the same section.**

Lines 255-257: *“The effects of oceanic heat entrainment and advection (Bushuk et al 2021; Holland et al 2013) might be more suited to explain the variability of this region, despite being located in the inner zone.”*

12. Also in Methods, you need to provide more information on the time periods for the various data sources for MLD and the gap-filling procedure.

Answer & Action:

Thanks for the suggestion. **We added more details on the product in the Method section.**

Lines 303-307 *“Conductivity-temperature-depth (CTD; 1970-2018), Argo floats (Argo international programme; 2000-2018) and marine mammal-borne sensor profiles (Marine Mammals Exploring the Oceans Pole to Pole programme; 2004-2018) were included. Generalized least squares linear-regressions of individual in situ profiles are performed around each grid point to produce gridded maps of climatological mean fields.”*

13. Methods – you do not state what data you have used for sea ice drift.

Answer & Action:

We indeed forgot the citations to the product (Kimura and Wakatsuchi 2011; Kimura et al. 2013). Apologies. **We now give more details on the ice drift data and added the corresponding references in the Method section**

Lines 343-344: *“[...] ice drift fields derived from AMSR-E brightness temperature by a cross-correlation algorithm (Kimura and Wakatsuchi 2011; Kimura et al. 2013).”*

14. Line 341 – “horizontal velocity”

Answer:

Our mistake. The corresponding sentence is no longer in the revised version.

15. Lines 345-365 – I am not 100% sure I understand what you did here. At first, it seemed you computed the mixed layer heat from the total fluxes, but nowhere do you explain how you computed the horizontal advective fluxes, the entrainment flux or the diffusive flux, or what values you use for the various coefficients. So it looks like you don't actually compute fluxes, but estimate if they are important for this broad-scale analysis by the fits in figure 3, and are only deriving a relationship between the flux, the MLH and the date of advance and retreat. This is not very clear, as elsewhere you describe using flux estimates (but for what, I am not sure). If I am correct, this description could be clearer if you simplified by deleting the equation on 338 and start from the equation on 343 (please number your equations!), and just explain that Q_t is the total net flux as you do, and that you are not attempting to estimate it. If I am wrong, and you do compute the fluxes, then you need to explain a lot better.

Answer & Action:

Thank you for the suggestion and apologies for the confusion. Please refer to the answer to your specific reviews no. 6 and 7.

Following your suggestion, we deleted the equation of former line 338 to clarify the methods.

16. Ref 30 – formatting issue here.

Answer & Action:

Our mistake. **We corrected the formatting issue.**

REVIEWERS' COMMENTS

Reviewer #1 (Remarks to the Author):

The authors were very thorough in their revisions. Figs S3 and 4 are nice new additions. I have a better understanding of the work now too, and have learned that only the SIC tendencies lag the date of ice advance. The tendencies are used to characterize the sea ice processes occurring in the inner/outer zones. I suppose it is reasonable to assume that these processes are essentially the same before and after ice advance. I accept that the authors don't want to analyze sea ice model results, even though I think it would strengthen their case for analyzing sea ice concentration tendencies after the date of advance.

I think there are enough gems in this paper to warrant publication. I think the partitioning into inner and outer zones will be beneficial for further process-level investigation and for climate model evaluation.

A few typos, etc:

Line 63 has an extra "not"

Line 65-68 consider adding a few words to say that T_h and D_y are proportions of the total sea ice concentration tendency.

Line 101 please define N_t^2 (it is also not defined in Fig S4). You might find the following paper a good one to review too

<http://dx.doi.org/10.1175/jpo-d-18-0184.1>

Line 238 The qualifying sentence about weak/insignificance in large parts of the seasonal ice zone confused me. Were the correlations just quoted excluding the seasonal ice zone? I didn't see any mention of a specific domain.

Line 348 I think this should read "represents the flux divergence of sea ice". It doesn't make very good sense the way it was written.

Line 357 I think (Delta sea ice concentration) -> (Delta SIC)

Line 358 I think D -> Dy and T -> Th

Drivers of Antarctic sea ice advance

Submitted to Nature Communications

Response to reviewers

K. Himmich et al.,

August 2023

Reviewer #1

General feedbacks:

The authors were very thorough in their revisions. Figs S3 and 4 are nice new additions. I have a better understanding of the work now too, and have learned that only the SIC tendencies lag the date of ice advance. The tendencies are used to characterize the sea ice processes occurring in the inner/outer zones. I suppose it is reasonable to assume that these processes are essentially the same before and after ice advance. I accept that the authors don't want to analyze sea ice model results, even though I think it would strengthen their case for analyzing sea ice concentration tendencies after the date of advance.

I think there are enough gems in this paper to warrant publication. I think the partitioning into inner and outer zones will be beneficial for further process-level investigation and for climate model evaluation.

Answer:

Thank you!

Specific comments:

Line 63 has an extra "not"

Answer & Action:

Thanks for spotting this mistake. **We removed the duplicate, line 62.**

Line 65-68 consider adding a few words to say that Th and Dy are proportions of the total sea ice concentration tendency.

Answer:

You are correct, our formulation was not clear enough.

Actions:

- We replaced “to the sea ice concentration budget” by “to the total sea ice concentration tendency”.
- **Lines 64-66:** “Instead, we evaluate the thermodynamic (*Th*) and dynamic (*Dy*) contributions to the total sea ice concentration tendency over the 30 days following *da* [...]”

Line 101 please define N_t^2 (it is also not defined in Fig S4). You might find the following paper a good one to review too

<http://dx.doi.org/10.1175/jpo-d-18-0184.1>

Answer:

We should have been more explicit on the definition of N_t^2 , you are right.

As for the Wilson et al. (2019) paper, it does give very useful insights on processes influencing the mixed layer stability in winter but does not look at summer processes, about which are we are the most interested.

Actions:

- We now only refer to the stability of at the base of the mixed layer in the main text.
- **Lines 91-92:** “[...] the temperature profile at the base of the mixed layer is thermally unstable in the outer zone during the first three months of the advance season [...]”
- We explicitly define N_t^2 in the analysis of Supplementary Figure 4:
- “ N_t^2 is proportional to the vertical temperature gradient at the base of the mixed layer:

$$N_t^2 = g\alpha \frac{\partial T_b}{\partial z}$$

where T_b is the temperature at the base of the mixed layer, α , the thermal expansion coefficient at constant pressure $\alpha = -\frac{1}{\rho} \frac{\partial \rho}{\partial T}$ and g , the gravity acceleration. $N_t^2 < 0$ ($N_t^2 > 0$) indicates a negative (positive) temperature gradient and an unstable (stable) temperature profile at the base of the mixed layer.”

Line 238 The qualifying sentence about weak/insignificance in large parts of the seasonal ice zone confused me. Were the correlations just quoted excluding the seasonal ice zone? I didn't see any mention of a specific domain.

Answer:

You are also correct; we were not specific enough on where correlations are significant or not according to Figure 4.

Actions:

- We added that correlations are significant in “large part of the seasonal ice zone”:

- **Lines 186-187** : *“Based on detrended time series over 1982-2018, we find significant and relatively strong negative links [...] in large parts of the seasonal ice zone.”*
- **We also specified in which regions correlations are weak and insignificant:**
- **Lines 190-191**: *“. However, those correlations are weak or statistically insignificant close to the seasonal ice zone edge but also in the most East Antarctic and Maud Rise sectors [...].”*

Line 348 I think this should read "represents the flux divergence of sea ice". It doesn't make very good sense the way it was written.

Answer & Action:

We rewrote the sentence for more clarity.

Lines 294-295: *“The ice concentration flux divergence represents the effects of advection and divergence of sea ice caused by ice drift.”*

Line 357 I think (Delta sea ice concentration) -> (Delta SIC)

Answer & Action:

Thanks for spotting this mistake. **We corrected it, line 302.**

Line 358 I think D -> Dy and T -> Th

Answer & Action:

Thanks for spotting this mistake. **We corrected it, line 303.**